# From weak to intense downslope winds: origin, interaction with boundary-layer turbulence and impact on $CO_2$ variability

Jon Ander Arrillaga[1], Carlos Yagüe[1], Carlos Román-Cascón[1,2], Mariano Sastre[1], Maria Antonia Jiménez[3], Gregorio Maqueda[1], and Jordi Vilà-Guerau de Arellano[4]

[1]Departamento de Física de la Tierra y Astrofísica, Universidad Complutense de Madrid, Spain
[2]Laboratoire d'Aérologie, University of Toulouse, CNRS, France
[3]Departament de Física, Universitat de les Illes Balears, Palma, Illes Balears, Spain
[4]Meteorology and Air Quality Group, Wageningen University, Netherlands

**Correspondence:** Jon A. Arrillaga (jonanarr@ucm.es)

**Abstract.** The interconnection of local downslope flows of different intensities with the turbulent characteristics and thermal structure of the atmospheric boundary layer (ABL) is investigated through observations. Measurements are carried out in a relatively flat area 2-km away from the steep slopes of the Guadarrama Mountain Range (central Iberian Peninsula). Forty thermally-driven downslope events are selected from an observational database spanning the 2017-summer period, by using an objective and systematic algorithm that accounts for a weak synoptic forcing and local downslope wind direction. We subsequently classify the downslope events into weak, moderate and intense, according to their maximum 6-m wind speed. This classification enables us to contrast their main differences regarding the driving mechanisms, associated ABL turbulence and thermal structure, and the major dynamical characteristics. We find that the strongest downslope flows ($U > 3.5$ m s$^{-1}$) develop when soil moisture is low ($< 0.07$ m$^3$ m$^{-3}$), and the synoptic wind not so weak ($3.5$ m s$^{-1} < V_{850} < 6$ m s$^{-1}$) and roughly parallel to the direction of the downslope flow. The latter adds an important dynamical input, which induces an early flow advection from the nearby steep slope, when the local thermal profile is not stable yet. Consequently, turbulence driven by the bulk shear increases up to friction velocity ($u_*$) $\simeq 1$ m s$^{-1}$, preventing the development of the surface-based thermal inversion, and giving rise to the so-called weakly stable boundary layer. On the contrary, when the dynamical input is absent, buoyancy acceleration drives the formation of a katabatic flow, which is weak ($U < 1.5$ m s$^{-1}$) and generally manifested in the form of a shallow jet below 3 m. The relative flatness of the area favours the formation of very stable boundary layers marked by very weak turbulence ($u_* < 0.1$ m s$^{-1}$). In between, moderate downslope flows show intermediate characteristics, depending on the strength of the dynamical input and the occasional interaction with downbasin winds. On the other hand, by inspecting individual weak and intense events, we further explore the impact of downslope flows on $CO_2$ variability. By relating the dynamics of the distinct turbulent regimes with the $CO_2$ budget, we are able to estimate the contribution of the different terms. For the intense event, indeed, we infer a horizontal transport of 67 ppm in 3 h driven by the strong downslope advection.

# 1  Introduction

Thermally-driven slope winds develop in mountainous areas, when the large-scale flow is weak and skies are clear, allowing greater incoming solar radiation during daytime and larger outgoing longwave radiation during the night (Zardi and Whiteman, 2013). Under this situation, wind direction is reversed twice per day: thermally-driven winds flow upslope during the day

and downslope during the night (Atkinson, 1981; Whiteman, 2000; Poulos and Zhong, 2008). The thermal disturbances that produce them have different scales and origin: from local hills and shallow slopes (Mahrt and Larsen, 1990) to large basins and valleys which extend horizontally up to hundreds of kilometers (Barry, 2008). The different scales, however, are not independent; for instance, local downslope flows converge at the bottom of the valleys generating larger-scale flows, which in turn influence the mountain-plain circulations.

Mountainous sites have struck the attention of many studies for plenty of reasons: in particular due to the influence on fog formation (Hang et al., 2016), diffusion of pollutants (Li et al., 2018), and also due to the important role that slope flows play in the thermal and dynamical structure of the Atmospheric Boundary Layer (ABL) and its morning and evening transitions (Whiteman, 1982; Sun et al., 2006; Lothon et al., 2014; Lehner et al., 2015; Román-Cascón et al., 2015; Jensen et al., 2017).

In this work, we focus on the locally-generated thermally-driven downslope winds (i.e. not driven by the basin or valley)

during nighttime, and the external factors and physical processes driving their formation and subsequent evolution. The latter is closely linked to the turbulent characteristics of the Stable Boundary Layer (SBL) and the variability of $CO_2$. Several external factors have been documented to affect these downslope winds: the steepness of the slope and the distance to the mountain range (Horst and Doran, 1986), the canopy layer (Sun et al., 2007), spatial variations in soil moisture (Banta and Gannon, 1995; Jensen et al., 2017), and the direction and intensity of the synoptic wind (Fitzjarrald, 1984; Doran, 1991). However, most

of the investigations analysing the influence of those external factors are carried out using numerical simulations, and there is a lack of observational studies to validate them. Oldroyd et al. (2016) stressed the importance of adequately describing the conditions under which pure thermally-driven or katabatic flows form, versus downslope winds which have a partial dynamical contribution (further described in Chrust et al. (2013)). From our observational analysis of several events we particularly focus on how soil moisture and the large-scale wind affect the onset time, nature and different intensities of thermally-driven

downslope winds, from an observational analysis of several events.

Soil moisture acts enhancing or reducing the thermal component, whereas the large-scale flow introduces a dynamical input which can considerably intensify downslope winds and modify their onset time. Banta and Gannon (1995) carried out numerical simulations using a two-dimensional model and found that katabatic flows are weaker over a moist slope than over a dry one. They quantified a greater downward longwave radiation and soil conductivity under moister conditions, which gives rise to a

reduced surface cooling. However, Jensen et al. (2017) found an opposite correlation, which they atributted to the sensitivity of the soil-moisture parametrisation in the numerical simulations. As a matter of fact, Sastre et al. (2015) showed with a numerical experiment at contrasting sites that soil moisture differences do not affect the afternoon and evening transition values with the same intensity, but depending on the site. With respect to the background flow, by using a one-dimensional model, Fitzjarrald (1984) observed that the onset time of katabatic winds is affected by the retarding effect of the opposing synoptic flow and

reduced cooling rates. Jiménez et al. (2019) found that moderate background winds in the direction of the thermally-driven flow enhance the latter by adding a dynamical component.

On the other hand, knowing whether a certain night the downslope flows will be weak or intense enables us to predict how turbulence in the SBL will behave. It is well known that under weak large-scale wind, turbulence is weak and patchy (Van de Wiel et al., 2003, 2012a; Mahrt, 2014), giving rise to the so-called very stable boundary layer (VSBL). On the contrary, when the large-scale wind is strong, shear production increases substantially and turbulence is continuous, producing near-neutral conditions in the SBL (Mahrt, 1998; Sun et al., 2012; Van de Wiel et al., 2012b), and the so-called weakly stable boundary layer (WSBL). Being able to foresee the occurrence of these two regimes has been in the eye of many studies, and some attempts have been made to characterise the transition between the regimes using diverse criteria such as the geostrophic wind (Van der Linden et al., 2017), local (Mahrt, 1998) and non-local scaling parameters (Van Hooijdonk et al., 2015), and the wind speed (Sun et al., 2012). More or less directly continuous turbulence in the SBL has been linked with a stronger background wind (e.g. > 5–7 m s$^{-1}$ in Van de Wiel et al. (2012b)) and low-level jets (Sun et al., 2012), or even with occasional irruption of sea-breeze fronts (Arrillaga et al., 2018). Our second aim is therefore to explore the direct implication of thermally-driven downslope winds generated by the presence of steep topography, in the occurrence of the two SBL regimes.

A relevant aspect of our site is its location in a relatively small flat area close to the mountain range. We therefore encounter a scenario different from other sites located at slopes where the SBL barely becomes very stable, since the shear production linked with the downslope wind is large and continuous, and buoyant turbulence production may occur even when the stable stratification is present (Oldroyd et al., 2016). In our site, however, VSBLs associated with relatively strong surface-based thermal inversions take place occasionally.

Connected also to the dynamics of the SBL, another relevant issue on this topic is the impact of downslope winds of different nature on the concentration of scalars of high relevance such as the $CO_2$. Previous studies have documented its influence in coastal areas (Cristofanelli et al., 2011; Legrand et al., 2016) and mountainous regions (Sun et al., 2007; Román-Cascón et al., 2019). Sun et al. (2007) found that downslope flows transported $CO_2$-rich air from the Rocky mountains, and Román-Cascón et al. (2019) observed that horizontal $CO_2$ advection can be relatively important over heterogeneous surfaces affected by different emission areas. Not only advection, but local turbulence fluxes can also be influenced: Sun et al. (2006) observed an anomalous positive $CO_2$ flux just after sunset, suggesting that it was due to the sudden transition from upslope to downslope flow. Being able to better quantify the influence of mesoscale flows on the $CO_2$ budget can help to reduce the large discrepancy from modelling studies in reproducing the land-atmosphere exchange for this gas (Rotach et al., 2014).

The aim of this work is to increase in the knowledge of:

[1] The external factors and physical processes that modulate the onset time and nature of thermally-driven downslope flows.

[2] The interaction of these downslope winds with local turbulence and the implication in the characteristics of the SBL.

[3] The role of advection and local turbulent fluxes, linked with the distinct downslope winds and the associated SBL regimes, in controlling $CO_2$ mixing ratios.

In order to shed light on those aspects, we perform an objective selection of downslope events and group them together according to their maximum wind speed, so that their intensity and the turbulent characteristics of the SBL are clearly associated.

The article is structured as follows. We detail the observational data employed and the objective criteria for selecting downslope events in Sect. 2. Section 3 addresses the main characteristics of these events, explores the influencing factors and the interaction with turbulence. We pursue the analysis of representative events focusing on the underlying physical mechanisms and particular characteristics in Sect. 4. Section 5 deepens the analysis by inspecting in detail the contribution of downslope flows to the variability of $CO_2$ concentrations. We finish with the relevant conclusions and two appendixes, which provide supplementary information about the footprint analysis and assessment of static stability in thermal profiles.

## 2 Data & Method

### 2.1 Site: La Herrería

The observational site employed in this work (meteorological, soil and $CO_2$ mixing ratio) is located beside *La Herrería* Forest (40.582° N, 4.137° W, 920 m asl), from which the name is adopted. La Herrería is placed at the foothill of the Guadarrama Mountain Range in central Spain, approximately 50 km NW of the city of Madrid (see Fig. 1a).

The site is placed at around 2 km from the steep slope of the Guadarrama mountain range (see Fig. 1b), which has an slope angle of around 25° in the main downslope direction (295°; approximately W-NW, from which the most intense downslope winds blow). The closest peak, Abantos, is 1763 m high, and the summit Peñalara is at 2420 m asl; both pointed in Fig. 1a.

The site is close to a highly vegetated area to the W, being also close enough to the small urban areas of San Lorenzo de El Escorial to the NW and El Escorial to the E-SE. In addition, it is relatively close to the large metropolitan area of Madrid, where concern regarding high pollution levels has increased in the last years (Borge et al., 2016, 2018). The diffusivity of pollutants is highly affected by the presence of downbasin winds in the city and surrounding areas blowing from the NE (Plaza et al., 1997), which develop from converging nocturnal flows at the centre of the basin. Besides, the generation of downbasin drainages causes fog formation in the centre of the Iberian Peninsula, affecting visibility, amongst others, in Madrid-Barajas Adolfo Suárez airport (Terradellas and Cano, 2007). Understanding the mechanisms that modulate downslope winds in this region is therefore of high importance.

The analysis is carried out during summer, a season that is characteristically very dry and warm and with nearly quiescent large-scale conditions in central Spain, even at the mountainous areas (Durán et al., 2013). The soil is particularly dessicated at the end of the season, which makes it different from other mountainous areas in Europe as the Pyrenees or the Alps. Summer 2017 was very warm and very humid (AEMET, 2017) in this region, following a very warm and very dry spring. In any case, it was not a particularly rainy season and in fact, precipitation during summer 2017 took place just over a few days, so that the dessicated soil experienced sharp moisture increases. This sets up a striking working frame to explore the role of soil moisture in the surface-energy balance and the associated consequences on the intensity and nature of downslope flows. And finally, the area surrounding the station is located in a relatively flat area (its slope angle is of around 2°) close to the Guadarrama mountain range (see Fig. 1b), which allows the formation of strong surface thermal inversions. These inversions are sporadically eroded by the drained downslope flows, providing an interesting scenario for the investigation of the distinct SBL regimes.

Regarding the vegetation and land use, the observational site is placed in a pasture grassland with scattered 3–5 m high shrubs and small trees, and the soil is composed of granite and gneiss. At around 2 km towards the SW, a broadleaved deciduous forest is found, and at the same distance to the NW, approximately where the steep slope starts, a mixture of needleleaved evergreen (coniferous) tree cover and mosaic-tree and herbaceous cover. During nighttime, the footprint of the fluxes measured at 8 m lies broadly within the fetch of the surrounding area. Nevertheless, the estimated footprint area can increase horizontally up to 150-200 m under very-stable conditions associated with the weakest downslope flows, inducing additional input from further inhomogeneities, although their contribution is generally small. An analysis of the calculated flux-footprint for three representative distinct downslope events is provided in Appendix A.

## 2.2 Data: Meteorological observations and post-processing

Standard meteorological measurements and eddy-covariance (EC) fluxes are recorded in the 10-m high fixed tower in La Herrería. La Herrería tower is part of the Guadarrama Monitoring Network (GuMNet, 2018), which aims at providing observational meteorological and climatological records to deepen into scientific research in the mountainous area of *Sierra de Guadarrama* (Durán et al., 2018). Data from aspirated thermometers, cup anemometers, radiometers, a wind vane and IRGASON devices among others are recorded along the mast. From the IRGASON equipment the three components of the wind, temperature and $CO_2$ measurements are obtained at high frequency (10 Hz), which allows the evaluation of different turbulent parameters from the EC technique. Measurements of the soil moisture are also taken.

For this study, measurements were carried out over an intensive campaign in Summer 2017 (22/06–26/09). Supplementary instruments were deployed along the mast for additional measurements, including *inter alia* an extra IRGASON and radiometer. Table 1 gathers specifications about the devices and the variables employed in this study. All the variables are averaged over 10 min.

Main correction and processing procedures applied to raw high-frequency time series are based on the software from the EasyFlux DL program (Campbell-Scientific, 2017), which provides fully corrected turbulent fluxes by applying some corrections frequently used in the related literature. The EasyFlux postprocessing software shows high correlation with the extensively used EddyPro software (Zhou et al., 2018). Despiking of the high-frequency time series are carried out using diverse diagnostic codes and signal strengths. Turbulent parameters inferred from IRGASON measurements were discarded when the signal strength was below 0.9, associated mostly with high values of the relative humidity and/or precipitation. Nevertheless, that situation was rarely observed within the analysed database, since the algorithm employed and described in the following section ensures fair-weather conditions. Frequency corrections are applied by using transfer functions for block averaging (Kaimal et al., 1989). An averaging time of 10 min was fixed for the calculation of turbulent parameters, which is considered standard for micrometeorological datasets (Mauritsen and Svensson, 2007), and appropriate during the afternoon and evening transition of the ABL. The election of the averaging window was supported by multi-resolution flux decomposition analyses for few representative events (not shown). On the other hand, the double-rotation method was applied to the sonic coordinate system (Kaimal and Finnigan, 1994), so that errors in the measurement of the turbulent parameters associated with alignment issues were corrected. Other data-quality control checks include moisture corrections to the sonic temperature following Schotanus

et al. (1983) to derive the air temperature and sensible heat flux, and air-density fluctuations were corrected by applying the WPL correction (Webb et al., 1980) to water-vapour and $CO_2$ turbulent fluxes. Additional minor corrections and quality-control checks are described in Campbell-Scientific (2017). Apart from those procedures, some other manual checks were considered. The results from the EasyFlux software were compared with the results obtained with our own programs, which also apply various postprocessing procedures. These, include quality-control checks and the rotation of the sonic-anemometer axes among others. The comparison showed high correlation and very good agreement.

## 2.3 Method: Downslope-detection criteria

The systematic and objective algorithm developed in Arrillaga et al. (2018) to detect sea-breeze events, and adapted to select mountain-breeze occurrences in three different areas in Román-Cascón et al. (2019), is used here. We adjust the algorithm for selecting events that fulfil predefined thermally-induced downslope criteria; i.e., when a shift of the wind direction is observed from the upslope to the downslope direction during the afternoon and evening transition, and always under a weak synoptic forcing. In this way, we evaluate the characteristics and impacts of the downslope flows in a more robust and objective way. Besides, the algorithm defines a benchmark which is the onset of the flow, enabling the clustering of different events and their analysis in a consistent way.

The algorithm consists of four different filters, which are shown in Table 2. The first three are coincident with the algorithm defined in Arrillaga et al. (2018), just modifying the precipitation-amount threshold for Filter 3: 0.5 instead of 0.1 mm/day, since weak and scattered showers (< 0.5 mm) do not alter the onset and development of downslope flows. From detailed individual analysis we found that the large-scale conditions favourable for the formation of thermally-driven downslope flows are those that apply for sea breezes, i.e. quiescent synoptic forcing and not having the passage of synoptic fronts (Filters 1 and 2 respectively). This fact supports the strong value of this method. The last filter (Filter 4) is based on specific criteria for downslope flows in La Herrería, and was defined after a thorough inspection of the wind behaviour around sunset on days passing the first three filters. An event is selected as 'downslope' when wind direction at 10 m is roughly perpendicular to the mountain-range axis (see Fig. 1a), i.e. within the range [250°–340°], for at least 2 h at a time between 1600 UTC and 2400 UTC (1800 and 0200 LT respectively). With this filter, events mainly driven by downbasin flows (from the NE) are rejected. In any case, downslope events are occasionally, and during short periods, interrupted by downbasin winds.

The downslope flow usually lasts until sunrise, when a strong veering of the wind direction occurs. However, in this study we just focus on the first stage of these flows, since the main objective is to investigate their connection with the onset of the SBL (defined as when the sensible heat flux, $H$, turns negative), and the different regimes associated. During the summer months, the onset of the downslope flow takes place usually before sunset, around 1800 UTC, but it can be considerably advanced or delayed depending on a number of factors, which are investigated in Sect. 3.2. A wide variability in the onset time of the katabatic flow was also reported in previous studies (Papadopoulos and Helmis, 1999; Pardyjak et al., 2009; Nadeau et al., 2013).

We identify the onset of the downslope flow as the first value within the 2-h range of continuous downslope direction. Having that onset time as a reference, we explore the characteristics and nature of downslope flows, their interaction with turbulence and the impact on $CO_2$ concentrations in the next sections.

## 3  Characteristics of the downslope flows

Forty thermally-driven downslope wind events were selected from the analysed period (94 days in total). The algorithm is very rigorous to ensure that the selected events are mainly thermally-driven downslope flows, since we just focus on the days with a weak large-scale wind in which there is a shift from the daytime upslope to the nighttime downslope wind direction. The results presented hereinafter are related to these 40 downslope events.

### 3.1  Wind direction and intensity

Figure 2 shows the direction and intensity of the selected events over the 2-h period subsequent to the onset of the downslope flow. The mean downslope direction is around 295° and the variation around this direction is small; the largest oscillations in the direction are observed for weak intensities. Within the database of the 40 events we find diverse cases depending on the maximum intensity of the downslope flow. We represent in Fig. 3 the vertical profile of the wind speed at the time of the downslope onset (a) and when the intensity of the flow is maximum at 6 m (b) by using box plots, which contain information about the frequency distribution at each observational level (3, 6 and 10 m). At the time of the onset the downslope flow is weak

at all levels (Fig. 3a); for instance the median at 10 m is slightly over 1 m s$^{-1}$. It can be noted that the median at 3 m is similar to that at 6 m, and the first quartile is even smaller at 6 m. This occurs because in some events prior to the identified onset at 10 m a very shallow katabatic or a skin flow is usually developed, and is only reflected at 3 m. This skin flow (also denominated katabatic flow, as for instance in Oldroyd et al. (2016)) can be observed when turbulence is very weak and thermal stratification is very stable (Mahrt et al., 2001; Soler et al., 2002; Román-Cascón et al., 2015), and occasionally gives rise to a greater wind

speed at 3 than at 6 m, and a few times even greater than at 10 m. However, when the downslope flow is more intense, wind speed increases with height within the 10-m layer from the surface (for instance note that the third quartile is greater at 6 than at 3 m). A maximum jet is probably found above 10 m, in accordance with the definition of *downslope flow* given in Oldroyd et al. (2016), but we do not have the measurements to check it.

From Fig. 3b, the intensity distribution at all levels can also be observed, but in this case when the maximum 6-m wind speed is recorded. We find in this case that the distribution above the median is elongated at all levels. In fact, the level of 6 m is employed to classify downslope events according to their maximum intensity and the associated erosion of the surface-based thermal inversion. The reasons for employing the level of 6 m are outlined below. Firstly, red crosses pinpoint an event identified as an outlier due to its high intensity at all levels (e.g. $U > 6$ m s$^{-1}$ at 10 m). Together with the wind maximum,

the surface thermal inversion is very weak or non-existent, and the maximum of turbulence measured from the turbulent kinetic energy ($TKE = [(1/2)(\overline{u'^2} + \overline{v'^2} + \overline{w'^2})]^{(1/2)}$) and friction velocity ($u_* = [(\overline{u'w'})^2 + (\overline{v'w'})^2]^{1/4}$) is even greater than the daytime maximum of the typical diurnal cycle (generally $u_{*,\,max} \simeq 0.5$–$0.7$ m s$^{-1}$). We find in addition two other events

with the above-mentioned features which are included within the right whisker. These three events are classified hereinafter as intense downslope events, and they all meet the criteria that the maximum 10-min wind speed at 6 m is greater than 3.5 m s$^{-1}$. It must be noted that this threshold is not very high, but in the context of a weak synoptic forcing and comparing to the rest of the events, we can consider them to be relatively intense. Secondly, in some events turbulence is very weak and the surface-based thermal inversion is not eroded ($u_* < 0.1$ m s$^{-1}$ mostly). They all occur when wind speed is very weak, and hence we classify as weak events (14 in total) those in which the maximum wind speed at 6 m is below 1.5 m s$^{-1}$. The characteristics of these weak downslope flows conform with the definition of pure thermally-driven downslope or katabatic flows, as defined in Oldroyd et al. (2016) or Grachev et al. (2016). Finally, the cases in which the maximum wind speed at 6 m is between 1.5 and 3.5 m s$^{-1}$ are classified as moderate downslope flows (23 in total). A summary of the classification is shown in Table 3. At the levels of 3 and 10 m the events showing different features cannot be so clearly detached, and therefore the level of 6 m is employed for the classification. Flocas et al. (1998) for instance studied katabatic flows at a similar height (7 m), since the influence of the large-scale wind was minimised at this level.

This classification is employed in the following sections to better illustrate the differences between the downslope events, their formation and development mechanisms, and the very distinct way in which they interact with local turbulence (in particular this is addressed in Sects. 4 and 5).

### 3.2 Factors influencing intensity

Once the downslope events are classified according to their maximum intensity, we first explore the factors that induce different intensities. Figure 4 shows a histogram with the difference between the onset time of the downslope flow and sunset time (it ranges from 1810 UTC in September to 1940 UTC in June), for different intensities in colours, and in lines for a different static stability of the thermal profile at the moment of the onset. Values of the net radiation ($R_n$) and TKE are also indicated. The static stability was estimated by fitting the virtual potential temperature at the different vertical levels (surface, 3, 6 and 10 m) to a logarithmic profile, as explained in detail in Appendix B. Virtual potential temperature was calculated using measurements from aspirated thermometers and a T/RH probe (see Table 1). The skin temperature was calculated from the upward longwave radiation by employing the Stefan-Boltzmann law.

From the classification introduced in Appendix B we infer the relationship between the earlier or later onset of the downslope wind, turbulence and the associated thermal stratification at the moment of its onset, and the intensity of the flow. On the one hand, intense downslope flows develop when the onset takes place prior to 1.5 h before sunset, the stratification within the first 10 m is still unstable, and $R_n$ and TKE are always greater than 80 W m$^{-2}$ and 0.8 m$^2$ s$^{-2}$ respectively. On the other, weak downslope or katabatic flows occur when the onset takes place later than 1.5 h before sunset and with neutral or stable stratification. In this case TKE is almost an order of magnitude smaller (in the order of or smaller than 0.1 m$^2$ s$^{-2}$), and $R_n <$ 40 W m$^{-2}$ always; it is in fact negative in more than 80 % of the cases. Moderate downslope flows, however, occur either when the onset takes place earlier or later, and hence independently of turbulence and the associated thermal stratification.

This result suggests the existence of external factors affecting the earlier or later onset of downslope winds. Particularly we focus on the soil moisture and the large-scale wind.

To explore the influence of soil moisture, we define an index that provides a measure of the relative dessication of the soil over the summer. This soil-moisture index is defined as the ratio between the observed liquid water volume and the maximum value throughout the analysed period (0.14 m$^3$ m$^{-3}$): $SM_i = SM/SM_{max}$. We separate the events into drier ($SM_i \leq 0.5$) and moister ($SM_i > 0.5$) cases. On the other hand, to explore the influence of the large-scale wind ($V_{850,18}$: at 850 hPa at 18 UTC) we separate the downslope events into very weak ($V_{850,18} \leq 3.5$ m s$^{-1}$) and weak ($V_{850,18} > 3.5$ m s$^{-1}$) synoptic forcing.

The influence of the large-scale wind speed and direction, and soil moisture, are investigated in Fig. 5. The maximum downslope intensity at 6 m together with the direction of the synoptic wind are represented in wind-rose form for the above-mentioned drier (a,b) and moister (c,d) cases, and for a very weak (a,c) and weak (b,d) synoptic forcing. The synoptic wind is estimated from the NCEP reanalysis wind speed employed in the selection algorithm, by choosing the grid point at 850 hPa closest to La Herrería (40.5° N, 4° W) at 18 UTC. At that point, the 850-hPa level is approximately at 800 m above ground level (agl), sufficiently close to the surface as to be representative of the synoptic wind at the surface level, and far enough as to be out of the influence of downslope winds.

We find that intense downslope flows (orange-reddish) develop when the soil is drier, the large-scale wind is weak and blowing from the N-NW (Fig. 5b). This direction is perpendicular to the mountain range axis (Fig. 1a), and approximately coincident with the downslope direction (Fig. 5). However, under those conditions we find weak downslope or katabatic flows (dark blue) when the large-scale wind blows from parallel or opposite directions (S or E-NE for instance), although most of the katabatics occur mainly when the soil is moister and the synoptic forcing is very weak (Fig. 5c). Katabatic flows establish primarily for W-SW and S-SE large-scale winds, but they can also occur for N-NW winds when their intensity is very weak (Fig. 5a). Overall, the intensity of downslope winds increases with decreasing soil moisture, and increasing synoptic forcing with a N-NW direction.

The role of the soil moisture in the surface energy balance, is in general, complex. The longwave-radiative loss and soil-heat flux show a peak when the relative dessication of the soil is large (not shown). However, after precipitation has occurred soil moisture increases considerably and the cooling of the soil can also be enhanced. This bimodal behaviour is in any case vague and ambiguous, and it must be supported by more conclusive observational results.

Overall, we find that the combination of low soil moisture, which enhances the thermal forcing, and the synoptic wind direction coincident with the downslope direction (N-NW), adding an important dynamical contribution, induces an earlier onset of downslope flows. The onset occurs when stratification is still unstable and convective turbulence relatively strong, which enables the development of intense downslope flows. If those conditions are not met, the onset occurs later when stratification is already neutral or stable, which limits the intensification of downslope flows. To understand the evolution of downslope flows after their onset we explore their interaction with turbulence in the next subsection.

### 3.3 Interconnection between downslope flows and turbulence

Thermal stratification and the associated turbulence at the moment of the onset modulate the intensity and subsequent development of the downslope flow. If the downslope flow arrives when the stratification is still unstable and the surface thermal inversion (hereinafter measured from $\Delta\theta_v$) is not formed yet, the downslope flow strengthens progressively. Later, due to the

radiative energy loss ($R_n < 0$) the stable stratification is already established ($\Delta\theta_v > 0$), and the katabatic (thermal) contribution of the downslope flow is enhanced. Given that the wind shear associated with the downslope flow is already high, the negative $H$ strengthens ($H < 0$) and after a while compensates the energy loss at the surface, impeding the development of the surface-based thermal inversion and inducing near-neutral stability conditions (Van de Wiel et al., 2012a). In that way, intense downslope flows give rise to a WSBL. On the other hand, without a clear dynamical contribution of the large-scale flow, the downslope flow is established later, when negative buoyancy of the air adjacent to the surface enhances the katabatic input. However, this katabatic flow is not intense enough as to increase turbulent mixing substantially. Besides, the increase of the downslope intensity and the associated wind shear are limited by the stable stratification itself. Thus, even the maximum sustainable heat flux does not compensate the radiative energy loss. Consequently, the bottom of the SBL cools down, which contributes to enhance the stable stratification. This positive feedback occurring under weak downslope flows suppresses turbulence and gives rise to a VSBL (Van de Wiel et al., 2012b). For moderate downslope flows, any of the two regimes can occur depending on the onset time and the dynamical contribution of the large-scale flow. These mechanisms are explored in the following sections.

Figure 6 shows the temperature stratification $\Delta\theta_v = \theta_v$ (10 m) - $\theta_v$ (3 m), the turbulence velocity scale $V_{TKE}$ and $H$ with respect to $U$ at 6 m. $V_{TKE}$ is calculated as the square root of the TKE. We just represent the 10-min average values in which the wind direction is downslope (until 2400 UTC) and $H$ is negative, so that the downslope flow is present (not for instance the downbasin wind) and the SBL is already established. By representing $\Delta\theta_v$ *vs* $U$ we find a contrasting relationship for weak and intense downslope flows (Fig. 6a). Katabatic flows give rise to a strongly stratified SBL ($\Delta\theta_v$ up to 2 K), whereas intense downslope flows are linked with very weak or almost non-existent surface-based thermal inversions. Interestingly, for a few 10-min values with similar wind speed ($U \simeq 1$–1.5 m s$^{-1}$), the thermal stratification is very different between weak and intense downslope flows, which suggests the existence of distinct regimes in the SBL. On the other hand, we find for very weak wind speed ($U < 1$ m s$^{-1}$) a large variability of thermal stratification, that occurs because around the onset of the downslope flow the thermal inversion is still absent or very weak in most of the cases. At the upper limit, however, $\Delta\theta_v$ tends to zero when wind speed increases its value. Moderate downslope flows show both types of behaviour, with the transition taking place for $U \simeq$ 1.5 m s$^{-1}$.

By representing the turbulence strength $V_{TKE}$ *vs* $U$ at 6 m we confirm the sharp transition for $U = 1.5$ m s$^{-1}$ (Fig. 6b). Katabatic flows are associated with very weak turbulence ($V_{TKE} < 0.5$ m s$^{-1}$) that hardly increases with wind speed, while turbulence for intense downslope flows is considerably greater and increases approximately at a linear rate with $U$. This behaviour was first observed in Sun et al. (2012), and defined as the HOckey-Stick Transition (HOST) in Sun et al. (2015). They identified three turbulence regimes in the SBL depending on the relationship between turbulence and wind speed. In Regime 1 turbulence is very weak and generated by local shear; in Regime 2 turbulence is strong and generated by the bulk shear $U/z$ (hence the linear relationship with wind speed); and finally, in Regime 3 turbulence is moderate and mainly generated by top-down turbulent events. The three regimes are pinpointed in Fig. 6b. The threshold value for the wind speed depends on height, and sets the value above which the abrupt transition from Regime 1 to Regime 2 occurs. Since in our case $z = 6$ m, it is indicated as $V_6$ in the figure. Katabatics are clearly associated with Regime 1 and intense downslope flows, instead, with Regime 2. Moderate

downslope flows can give rise to either the three of the regimes. We just show the relationship for $z = 6$ m, and HOST in our case occurs for $V_6 = 1.5$ m s$^{-1}$, which is significantly lower than the value at that height from Sun et al. (2012) ($\simeq 3$ m s$^{-1}$) over relatively flat and homogeneous terrain (Poulos et al., 2002). This lower threshold could be partly induced by the proximity of the jet maximum, as for instance is shown in Fig. 8). Besides, $V_6$ coincides with the threshold value we anticipated for defining katabatic flows (Table 3). We measure a slope of $\sim$0.5 for $VTKE$ vs $U$ for regime 2, while is of $\sim$0.25 in the results from Sun et al. (2012).

In Fig. 6c we explore how downslope intensities and turbulence strength are manifested in terms of heat flux. The downward $H$, i.e. when the SBL is already established ($H < 0$), is represented in absolute values. The smallest $H$ values are observed for katabatic flows, when $U < 1$ m s$^{-1}$. The highest values take place for moderate downslope flows when $U$ lies between 1.5–2.5 m s$^{-1}$. We find a few data from some intense downslope flows in which $H$ is nearly zero for that wind-speed range, when the thermal inversion has just been formed. For intense downslope winds the peak is reached at $U \simeq 3$ m s$^{-1}$, and above that intensity $H$ decreases, since it is limited by the neutral stratification.

So far we have explored all the selected downslope events together, and learnt about their main characteristics, influencing factors and connection with the SBL regimes. We have observed how the turbulent characteristics of the ABL and the nocturnal regime can be predicted from the intensity of the downslope flow at 6 m. However, in order to better understand the mechanisms underlying their formation, development and their complex interaction with turbulence in the SBL, we target the analysis of individual representative events.

## 4    Analysis of representative downslope events

We choose representative weak, moderate and intense downslope events, so that their contrasting features and distinct influencing mechanisms are revealed. The weak downslope event is August 13 and is characterised by the presence of a katabatic flow and a VSBL. In short, the synoptic situation was marked by the Azores high and a thermal low over the Iberian Peninsula inducing a weak S flow, which was related to a weaker downslope intensity in Fig. 5. The $SM_i$ was high for this case. The moderate event is July 25, and this day the synoptic situation was also characterised by the Azores high, but in this case with a weak NE forcing, coinciding with the direction of the downbasin winds in the basin in which La Herrería is located. Besides, the soil moisture was low. This event was marked by the interaction between local downslope and downbasin winds. Finally, the intense event is July 27, again marked by the Azores high, but with a weak NW forcing coinciding with the direction of downslope flows, adding a dynamical input to the local flow. Soil moisture was also low for this case. In addition to the strong turbulence associated with the intense downslope flow, this event is chosen due to its particular influence on the $CO_2$ transport, which is addressed in Sect. 5.

## 4.1    Origin and underlying physical mechanisms

In previous sections we have shown that external factors such as soil moisture and the synoptic wind modulate the intensity of downslope winds, which is at the same time linked with an earlier or later onset time. In order to understand the physical

processes underlying the formation and subsequent evolution of the downslope flows, the momentum and heat budgets for the three individual events are explored following the two-dimensional simplified equations from Manins and Sawford (1979), which were employed in the recent studies from Nadeau et al. (2013) and Jensen et al. (2017). The slope-parallel momentum and heat budgets are respectively given by:

$$\frac{\partial \overline{u}}{\partial t} + \overline{u}\frac{\partial \overline{u}}{\partial s} + \overline{w}\frac{\partial \overline{u}}{\partial n} - \frac{g\overline{d}sin\alpha}{\theta_{va}} + \frac{\partial \overline{u'w'}}{\partial n} + \frac{1}{\overline{\rho}}\frac{\partial(\overline{p}-\overline{p_a})}{\partial s} = 0,$$

$$(I) \quad (II) \quad (III) \quad (IV) \quad (V) \quad (VI)$$

(1)

$$\frac{\partial \overline{\theta_v}}{\partial t} + \overline{u}\frac{\partial \overline{\theta_v}}{\partial s} + \overline{w}\frac{\partial \overline{\theta_v}}{\partial n} + \frac{\partial \overline{w'\theta_v'}}{\partial n} + \frac{1}{\overline{\rho}c_p}\frac{\partial \overline{R_n}}{\partial n} = 0,$$

$$(I) \quad (II) \quad (III) \quad (IV) \quad (V)$$

(2)

where $s$ and $n$ are the slope-parallel and slope-normal coordinates in accordance with the double-rotation of the axes, $t$ is time. $u$ is the streamwise wind, $w$ the slope-normal wind, $d = \theta_v - \theta_{va}$ is the temperature deficit defined as the difference between the perturbed vitual potential temperature and the unperturbed potential temperature, $\alpha$ ($\simeq 2°$) is the slope angle, g = (9.81 m$^2$ s$^{-2}$) gravity acceleration, $\rho$ the air density, $p$ is the measured local pressure, $p_a$ the ambient pressure field and c$_p$ (= 1005 J kg$^{-1}$ K$^{-1}$) the specific heat of the air at constant pressure. The bar is used for indicating 10-min time averaging, and the primes indicate a perturbation from the temporal mean.

Term I in Eq. 1 represents momentum storage, terms II and III represent horizontal and vertical advection respectively, term IV is buoyancy acceleration, term V the momentum turbulent flux divergence and term VI represents the along-slope pressure gradient. Regarding the heat-budget equation (Eq. 2), term I is the heat storage, terms II and III horizontal and vertical advection respectively, term IV the kinematic heat flux divergence and the last term represents $R_n$ divergence. For the calculation of $\overline{d}$, the difference between the levels of 2 and 20 m is calculated (as in Jensen et al. (2017)). For that, the values are obtained from the vertical extrapolation from the fit to the logarithmic profile (see Eq. B1). All derivatives are evaluated using forward finite differences, between the levels of 4 and 8 m in term V in Eq. 1 and term IV in Eq. 2, and between the levels of 1 and 2 m in term V in Eq. 2. On the other hand, the terms with slope-parallel gradients are considered residuals of the equations due to the lack of additional spatial measurements, and vertical advection in both equations (term III) is null ($\overline{w} = 0$) due to the imposition from the double-rotation method. 10-min averages are used.

The times series of these terms are represented and compared in Fig. 7 for the representative weak, moderate and intense downslope events, from 1600 UTC until 2400 UTC. Due to the lack of spatial gradients, the estimation of the residual has some uncertainty in particular for the momentum budget, since there are two terms within the residual of Eq.1 and only one in Eq. 2. In any case, the comparison between the three events explains the differences in the physical mechanisms underlying the formation and development of the distinct downslope flows. While the vertical axis for the heat budget is kept constant and is limited to 0.02 K s$^{-1}$ for better showing the differences, it is variable for the momentum budget due to the highly distinct relative weight of the different terms.

Prior to the onset of the weak downslope flow (Figs. 7a and 7b), the bottom of the surface layer cools down driven mainly by the kinematic heat flux divergence, as observed in the analysis from Jensen et al. (2017), whereas momentum is produced by the turbulent-flux divergence. This suggests the formation of a very shallow flow below the lowest observational level before the recorded onset. After the onset, the main source of momentum is buoyancy acceleration driven by the local surface cooling, in principle balanced by large-scale pressure and advective terms. This mechanism conforms also with the findings from Nadeau et al. (2013), confirming that weak downslope winds share the characteristics of katabatic flows. The cooling after the onset is primarily ruled by the heat-flux divergence, compensated by the heating produced by the radiative divergence. Despite the radiometers had previously been calibrated by being compared at the same level, the calculation of the divergence can be subject to errors due to the closeness (= 1 m) between the devices. Furthermore, Steeneveld et al. (2010) eliminated the radiative measurements coinciding with wind speed at 10 m below 1.5 m s$^{-1}$, to assure that radiometers were sufficiently ventilated. Hence, there might be certain uncertainty in the estimation of the radiative divergence, and the advective term could be responsible for balancing the cooling otherwise. In addition, this event shows a sudden turbulent burst at around 2400 UTC (which can be observed in Fig. 8a). It produces an upward momentum transport from the skin flow and a sudden warming due to turbulent mixing, demonstrating that the very stable regime may be occasionally interrupted.

The onset of the moderate downslope event (Figs. 7c and 7d) takes place rather earlier. According to the momentum budget, either advection or the along-slope pressure gradient, or both, are responsible for the intensification of the downslope flow when the buoyancy acceleration is still negative. When $R_n$ turns negative the downbasin wind arrives, and due to the collision between both, turbulence decreases and the further intensification of the downslope flow is prevented. Due to the interaction between the downbasin wind (NE) and the downslope flow (NW), turbulence is intermittent and of variable intensity, and the momentum balance is marked by the oscillating equilibrium between the momentum-flux divergence and the advective and pressure-gradient terms. Regarding the heat budget, the main source of cooling is advection, even after $R_n$ turns negative. After 2000 UTC, however, the advection produces warming and compensates the cooling from the heat-flux divergence.

During the intense event (Figs. 7e and 7f), the governing forcing in driving the intensification of the downslope flow is dynamical: either or both advective and large-scale pressure terms dominate over the local buoyancy acceleration, and are balanced by the turbulent-flux divergence. Given that the slope of the nearby mountain towards the NW is steep ($\simeq 25°$), and with an enhanced temperature deficit due to the greater dessication of the soil, the buoyancy-acceleration term is considerably greater over the slope than in La Herrería. Furthermore, the mountain-range axis is directed SW-NE, so that sunset takes place at the back of it. Therefore, the cooling starts before along the slope than in La Herrería. The heat budget shows that before $R_n$ turns negative, the source of cooling is advection from the slope, compensating the radiative heating. These findings confirm that the origin of the early flow is the advected cold drainage front from the nearby steep slope. Later, the main sources of cooling are the heat-flux divergence and the radiative divergence from 2100 UTC on, which are compensated by the warm advection. In any case, the cooling from the heat-flux divergence is smaller than in the moderate event, due to the constrain from the weak thermal stratification. A similar situation was reported in Papadopoulos and Helmis (1999), where the flow reached the foot of the slope as a drainage front from more elevated cold-air sources, before the establishment of the thermal inversion. This earlier onset at the foot was observed when relative humidity was lower at the slope site, when the local thermal structure

did not sustain katabatic motion, as occurs in our intense downslope event. They showed how advective effects dominated in this case and the arrival of the drainage front produced a temperature fall. Additionally, going along with our findings, they observed that the growth rate of this downslope flow was more intense due to weaker ambient stability, whereas the existence of a surface-based thermal inversion opposed the flow progression.

## 4.2 Nature and interaction with turbulence

After inspecting the physical mechanisms underlying the different evolution of the three representative downslope events, we characterise the associated wind and thermal profiles, as well as the interconnection of the flow with local turbulence, in Figs. 8, 9 and 10, for the weak, moderate and intense downslope events respectively. We show the time series of thermal stratification and friction velocity at 8 m in (a). Vertical profiles of the wind speed (b), $\theta_v$ (c) and heat and momentum horizontal turbulent fluxes (d) are shown below at three times of interest: at the moment of the onset, when $R_n$ turns negative and finally at 2230 UTC when the SBL is already well formed. The logarithmic fitting of the discrete observed measurements for both wind speed and $\theta_v$ is also depicted in (b) and (c). With respect to the turbulent fluxes, we can infer from their sign whether the jet associated with the downslope flow is located below or above the measurements (Grachev et al., 2016; Oldroyd et al., 2016). At the height of the wind-speed maximum both turbulent fluxes become zero, and above the jet $\overline{u'\theta_v'} < 0$ and $\overline{u'w'} > 0$, whereas below the jet $\overline{u'\theta_v'} > 0$ and $\overline{u'w'} < 0$.

Prior to the onset of the weak downslope or katabatic flow, the thermal stratification turns positive (Fig. 8), which induces the increase of the buoyancy-acceleration term as shown in Fig. 7. However, the local slope is shallow, and in the absence of a dynamical contribution from the nearby steep slope, the katabatic flow and the associated turbulence are very weak. $u_*$ is maintained below 0.1 m s$^{-1}$ throughout most of the event, revealing the presence of the VSBL. In fact, the thermal inversion strengthens up to 1.5–2 K until around 2100 UTC. As commented before, there is just a turbulent burst at around 2400 UTC, identified by an increase in $u_*$ of up to 0.3 m s$^{-1}$. The wind profile at the moment of the onset shows a jet below 3 m associated with a skin flow ($U < 1$ m s$^{-1}$), which is even sharper at 2230 UTC. The stronger stability is also reflected in the thermal profile at that time. At 2230 UTC, $\overline{u'w'}$ and $\overline{u'\theta_v'}$ become zero below 4 m (in particular the second), corroborating the presence of the katabatic jet or skin flow. At the moment of the onset, however, $\overline{u'w'}$ is slightly negative, suggesting the potential existence of an additional jet above 10 m, probably associated with the downbasin flow.

The moderate downslope wind is established when the thermal profile is still unstable (Fig. 9). When $R_n$ turns negative and the thermal inversion starts to develop, turbulence is still strong ($u_* = 0.5$ m s$^{-1}$). Subsequently, however, the arrival of the downbasin wind and the produced collision, reduces turbulence. Throughout the night, turbulence and the thermal inversion have an intermittent behaviour due to the alternation between the downslope and downbasin flows. Note that for this event 2300 UTC is chosen as the reference time instead of 2230 UTC, since at the latter time the downbasin wind is blowing. The form of the wind profile is rather homogeneous during the event (all the times shown are during the downslope stage), even under contrasting thermal profiles. It shows the presence of a possible jet below 3 m, although the profile is rather homogenised. $\overline{u'\theta_v'}$ is maintained negative at all times but the tendency with height is variable, as well as in the case of $\overline{u'w'}$, which turns positive

throughout the night. This complex behaviour could be explained by the alternation between the downslope and downbasin flows, and the presence of a subtle jet below 3 m, and another jet above 10 m.

The onset of the intense downslope wind takes place around 1640 UTC (Fig. 10), during the afternoon transition of the ABL. After this time, $u_*$ continues increasing slowly up to 0.5–0.6 m s$^{-1}$, and even up to 0.9 m s$^{-1}$ a few hours later, exceeding the value associated with the diurnal peak (not shown). As occurs with the moderate downslope wind, due to the dynamical input from the steep slope towards the NW, the intense downslope wind arrives when the stratification is still unstable. After $R_n$ turns negative slightly before 1900 UTC, $\Delta\theta_v$ is always maintained below 0.2–0.3 K due to the strong turbulent mixing. The wind profile is different from both the moderate and weak events: wind speed increases linearly with height between 3 and 10 m and intensifies throughout the night, exceeding 2 m s$^{-1}$ at 6 m when $R_n$ turns negative, and 3 m s$^{-1}$ at 2230 UTC. After stable conditions are established, the layer between 3 and 10 m is well mixed ($\Delta\theta_v \simeq 0$) despite the fact that the surface is already cooling down ($\theta_s < \theta_3$). The thermal inversion is limited to a thin layer close to the surface (below 3 m). This indicates the presence of near-neutral conditions and the set up of the WSBL. At the moment of the onset, the heat and momentum turbulent fluxes become near zero close to 4 m, indicating the possible presence of a low-level jet. Nevertheless, $\overline{u'w'}$ becomes more negative and $\overline{u'\theta_v'}$ more positive throughout the night, confirming the presence of the jet above 10 m.

## 5  Impact on CO$_2$ variability

We finally explore the impact of downslope flows of different intensities and the associated different turbulent characteristics on a relevant scalar: CO$_2$. For this analysis, the weak and intense downslope events are solely compared, since the budget for the moderate event is more complex due to the alternation of the downslope and downbasin winds.

The mixing ratio of CO$_2$ and the vertical turbulent fluxes are represented in Fig. 11 for the weak and intense downslope events. The CO$_2$ mixing ratio is normalised with respect to the daily mean. By doing so, we aim to reduce the uncertainty due to possible biases. In Fig. 11b the measured turbulent fluxes at 4 and 8 m, and the estimated soil respiration flux ($R_s$) are represented. Since the soil respiration is an important CO$_2$ source term near the surface, we decided to include it in the analysis. It has been calculated following Lloyd and Taylor (1994) and Jacobs et al. (2007a):

$$R_s = R_{10}\Big(1 - f(SM)\Big)\left(exp\Big(\frac{E_0}{283.15R^*}\Big)\Big(1 - \frac{283.15}{T_s + 273.15}\Big)\right), \tag{3}$$

where R$^*$ = 8.31·10$^{-3}$ kJ K$^{-1}$ mol$^{-1}$ is the universal gas constant, E$_0$ is the activation energy (we employ a value of 53.30 kJ mol$^{-1}$) and $T_s$ is the soil temperature, which has been estimated from the upward longwave radiation. R$_{10}$ is the reference soil-respiration value at 10° C under no water-stress condition, and it can vary significantly from site to site (Jacobs et al., 2007b). In this case, given the dry-soil conditions, we consider a value of R$_{10}$ = 0.10 $\pm$ 0.02 mg m$^{-2}$ s$^{-1}$. Finally, (1 - $f(SM)$) is a water-stress correction (Jacobs et al., 2007a) where:

$$f(SM) = C\frac{SM_{max}}{SM + SM_{min}}, \tag{4}$$

with C(=0.0016) being a constant, and *SM* the observed soil moisture at 4-cm depth. $SM_{max}$ and $SM_{min}$ are the respective recorded maximum (= 0.14 m$^3$ m$^{-3}$) and minimum (= 0.01 m$^3$ m$^{-3}$) soil-moisture values throughout the summer.

The normalised $CO_2$ mixing ratio at 4 and 8 m for the weak and intense events has the same values at 1600 UTC before the surface thermal inversion is set up (Fig. 11). It is therefore of great interest comparing these particular weak and intense events having the same initial mixing ratios but contrasting subsequent dynamical and stability conditions. During the weak event, the $CO_2$ starts increasing at around 1730 UTC when the turbulent fluxes at 4 and 8 m become positive. Slightly after the onset, which coincides with the set up of the SBL as reported in Sect. 4.2, the $CO_2$ starts to accumulate close to the surface until 2000 UTC approximately, due to the dominance of the soil flux over photosynthesis and dynamic transport. Later on, the balance between the divergence of the turbulent fluxes and the horizontal and transport explains the variability of the scalar.

The $CO_2$ for the intense downslope event shows a contrasting evolution. The onset occurs almost 2 h before the weak event, and the diurnal positive vertical $CO_2$ gradient is reduced beforehand. Due to strong turbulence, the vertical $CO_2$ fluxes reach values of up to 0.2–0.3 mg m$^{-2}$ s$^{-1}$ slightly after the establishment of the SBL at around 1900 UTC (see Fig. 10), following closely the estimated values of the soil respiration. From that moment and until 2200 UTC, the vertical gradient of the $CO_2$ is almost null, but the concentration increases around 6 ppm at both levels (note that it cannot be inferred from the normalised concentration in Fig. 11a). Considering in addition that the divergence of the vertical fluxes is mostly positive in that time range ($(\overline{w'CO_2'})_{8m} > (\overline{w'CO_2'})_{4m}$), the increase of the $CO_2$ concentration is explained by the non-local horizontal transport associated with the intense downslope flow. From the null vertical gradient of $CO_2$ between 4 and 8 m, and of $\theta_v$ between 3 and 10 m (see Fig. 10c), we can assume that the layer between 4 and 8 is well mixed. Furthermore, since *w* close to the ground is nearly zero, vertical advection can be neglected. After those assumptions, we can infer the horizontal transport in that 4-m layer between 1900 UTC and 2200 UTC, following the methodology from Casso-Torralba et al. (2008) based on well-mixed layers.

We based our analysis on the one-dimensional governing equation for $CO_2$ in between 4 and 8 m. We therefore neglected the effects of soil $CO_2$. In short, after applying the Reynolds decomposition and averaging of the velocity fluctuations, and by considering the continuity equation, we get Eq. 5 for the average $CO_2$ in the layer between 4 and 8 m. The wind has been aligned in the main component. We additionally neglect the horizontal turbulent flux divergence, since for this event the flux-fetch condition in the downslope direction is met at 8 m (see Fig. A1).

$$\frac{\partial \overline{CO_2}}{\partial t} + \overline{u}\frac{\partial \overline{CO_2}}{\partial s} + \frac{\partial \overline{w'CO_2'}}{\partial n} = 0. \tag{5}$$

The first term represents the storage, the second term horizontal advection and the third term, the divergence of the turbulent fluxes in the layer between 4 and 8 m. Since the layer is well mixed, we assume linearity of the turbulent fluxes with height

(Casso-Torralba et al., 2008). Integrating Eq. 5 in time we get to the following equivalence for the horizontal-transport term:

$$\int\limits_{1900}^{2200} \overline{u}\frac{\partial \overline{[CO_2]}}{\partial s}dt = -\int\limits_{1900}^{2200} \frac{\partial \overline{[CO_2]}}{\partial t}dt \; - \int\limits_{1900}^{2200} \frac{\Delta \overline{w'CO_2'}}{\Delta n}dt \tag{6}$$

The time evolution of the three terms is shown in Fig. 12, together with the wind-speed evolution. Since the horizontal gradient of the $CO_2$ concentration increases upwind of the downslope flow, the sign of the advective term is negative (note that it is represented in absolute values). After the corresponding calculations in Eq. 6, we obtain that a horizontal transport of 67 ppm over 3 h induced by the intense downslope flow compensates for the loss due to the vertical divergence (around 61 ppm in 3 h), resulting in an increase of the $CO_2$ storage of 6 ppm. This positive $CO_2$ advection is probably induced by the presence of a land use composed of forest, mosaic trees and shrubs upwind, towards the downslope direction. Given the increased plant respiration and soil flux, greater $CO_2$ concentrations are accumulated close to the surface during the night. Intense downslope flows, as demonstrated in previous sections, induce strong wind shear and considerable mixing of the lower SBL, which together with the strong flow, contribute to cause important transport of scalars such as the $CO_2$.

## 6 Summary and conclusions

Forty downslope events of different intensities, and with contrasting impacts on the turbulent characteristics of the SBL and $CO_2$ variability, were investigated. Measurements were carried out in an approximately flat area at the foothill of a high mountain range in central Iberian Peninsula during summer 2017. During that period, the soil underwent relatively strong dessication alterations. Observations of turbulent fluxes, $CO_2$ mixing ratios and other meteorological variables were recorded at various vertical levels along a 10-m tower. A systematic algorithm was employed in order to select unambiguously thermally-driven downslope winds, by using objective filters to account for large-scale and local forcings.

The selected downslope events were classified into three groups according to the observed maximum 6-m wind speed until midnight: weak, moderate and intense. By clustering them into these three groups, we were able to explore the factors that produce different intensities and their relationship with the different turbulent patterns in the SBL. Moreover, by exploring individual representative events, we investigated the mechanisms underlying their formation and development, and their specific thermal and dynamical features.

Weak downslope flows have maximum 6-m wind speeds below 1.5 m s$^{-1}$. They particularly take place when the large-scale wind opposes the downslope flow and soil moisture is greater than the summer median, which in general induces a smaller radiative cooling. These factors give rise to the formation of a prototypical katabatic flow, driven mainly by the buoyancy acceleration. Since the local slope is gentle, and in the absence of a clear dynamical input, their intensity is very weak. The katabatic jet is located below 3 m, and the associated turbulence is very weak. A positive feedback between the weak turbulence and the progressive cooling of the surface, which induces a more stable stratification, explains the formation of the very stable regime. Intermittent but weak turbulence can sporadically be observed.

Intense downslope flows have a maximum 6-m wind speed that exceeds 3.5 m s$^{-1}$. They mostly take place when soil moisture is lower than the summer median, the large-scale wind blows approximately in the downslope direction and its speed is greater than 3.5 m s$^{-1}$ at 850 hPa. These factors add an important dynamical input in the form of an advection from the nearby steep slope. It induces an early onset, when the thermal stratification is not stable yet, and an approximately linear wind profile above 3 m. The jet in this case is located above 10 m. Therefore, wind shear associated with the flow increases without being damped by the stable stratification itself. By the time the SBL is formed, the layer between 3 and 10 m is maintained well mixed and the thermal inversion is limited to a very thin layer close to the surface, reducing the thermal contribution of the katabatic flow. In this way, near-neutral conditions are reached and the WSBL is established.

Moderate downslope flows lie between weak and intense downslope flows, and show intermediate characteristics. Depending on the relative weight of the dynamical contribution to the flow and the interaction between local downslope and downbasin flows, their characteristics may be closer to either weak or intense downslope winds. Their impact on the turbulent characteristics of the SBL strongly depends on the maximum flow intensity that is reached. We found, indeed, that the wind speed is the most precise variable for representing the nocturnal regime transition: above a 6-m wind speed of 1.5 m s$^{-1}$, it is the bulk shear which dominates the turbulence production, and the thermal inversion is eroded significantly, giving rise to a regime transition.

Finally we inspected the impact of contrasting weak and intense downslope events on the $CO_2$ budget. For the weak event, minimal turbulence levels contribute to the accumulation of $CO_2$ close to the surface, and its concentration is sensitive to slight changes in the turbulent fluxes. For the intense event, instead, turbulence is considerably greater and consequently the layer between 4 and 8 m is well mixed. Under these conditions, we estimated the contribution of the horizontal transport in the downslope direction, which is of around 67 ppm in 3 h, contributing to the increase in storage of this scalar.

To sum up, we have assessed the main external factors and physical mechanisms underlying the formation and development of thermally-driven downslope winds of different intensities. By measuring the maximum intensity of downslope flows we are able to diagnose the turbulent characteristics of the SBL during the night. Being able to predict the influencing external factors more precisely is therefore of high interest to better forecast the nighttime turbulence and regime transition. Further observational investigations of the influence of soil moisture in the surface energy balance are needed. On the other hand, sudden turbulent bursts and collapses, and the interaction with gravity waves have not been explored in this work, which can also be relevant in producing perturbations in the SBL and regime transitions. Future studies should tackle with those features and a better performance of numerical models in reproducing them over complex terrain.

*Data availability.* Original data are freely available upon request through the GuMNet web: https://www.ucm.es/gumnet/.

**Appendix A: Footprint estimation**

The footprint of the turbulent fluxes is estimated by using the approximate analytical model from Hsieh et al. (2000), which is based on a combination of results from lagrangian stochastic dispersion models and dimensional analysis. This footprint model

was chosen in this work because it is developed for thermally stratified surface layers, it is practical and has been applied in many studies, giving satisfactory results when compared with other footprint models.

The cross-wind integrated footprint has the following form:

$$F^y(x, z_m) = \frac{1}{k^2 x^2} D z_u^P |L|^{1-P} e^{-\frac{D z_u^P |L|^{1-P}}{k^2 x}}, \tag{A1}$$

where $z_u$ is a length scale, $z_m$ the effective height of the sensors, D and P similarity constants which depend on the thermal stratification, $L$ the Obukhov length and k the Von Kármán constant. The footprint estimation is extended to 2D by adding the contribution of lateral spread assuming that is Gaussian, based on the formulation from Detto et al. (2006):

$$f(x, y, z_m) = \frac{1}{\sqrt{2\pi}\sigma_y} e^{-\frac{1}{2}\left(\frac{y}{\sigma_y}\right)^2} F^y(x, z_m), \tag{A2}$$

with $\sigma_y$ being the standard deviation of the lateral wind fluctuations.

By considering an approximate average height of the surrounding trees of $h = 5$ m, the footprint function is calculated and represented for the three representative events of the weak, moderate and intense downslope types in Fig. A1. It is depicted in the main downslope direction when the downslope wind is already developed (at 2100 UTC), superimposed on the map of the surrounding area from La Herrería. It is just shown for the sensor height of 8 m, since for 4 m the footprint area is smaller and hence the flux-fetch condition more easily fulfilled.

Most of the footprint area, delimited by the large black curve which corresponds to 90% of the total measured flux, is only affected by sparse bushes and short-medium trees of up to 5 m (the tallest ones located at 30-40 m towards the W-SW), so that the flux-fetch requirement is fulfilled. For the intense downslope event, the footprint is even unaffected by those inhomogeneities. Just in the weak downslope flow the footprint might be slightly affected by the tall poplars located towards the N, even though its input is anyway small.

## Appendix B: Assessment of the thermal profile

The thermal profile within the first 10 m over the surface is fitted to a logarithmic profile by the method of least squares. The fit is carried out employing four vertical values of $\theta_v$ calculated from measurements in La Herrería at the surface, and at 3, 6 and 10 m. The skin temperature is calculated from the upward longwave radiation. The fitting is carried out as follows:

$$\theta_v(z, t) = \alpha(t) + \beta(t)\ln(z) + \gamma(t)\ln^2(z) + \delta(t)\ln^3(z). \tag{B1}$$

We find that from the value of $\delta(t)$, which provides information about the curvature of the profile in its lowest part, we can infer the static stability of the thermal profile. We illustrate the thermal profile at three different times of day in Fig. B1, each time associated with a different static stability for a moderate downslope event: 25 July. The use of four vertical levels allows the fit to a cubic polynomial, and as a consequence more realistic near-surface profiles. Given also the limited number of measurements, the fit is exact (note that $r^2 = 1$), which would not occur in case of having a greater number of vertical measurements.

From a careful inspection of $\delta(t)$ for different events at various times of day and employing $\theta_v$ in K and $z$ in m, we classify the static stability of the lowest 10-m atmospheric layer into the three subsequent groups according to the value of $\delta(t)$:

$$Unstable: \quad \delta(t) < -0.3,$$
$$Neutral: \quad -0.3 \leq \delta(t) \leq 0,$$
$$Stable: \quad \delta(t) > 0.$$

(B2)

*Author contributions.* CY, MS, CRC, GM and JAA assisted with the collection and validation of the quality data, CY lead the project and conducted the field experiment, CRC and MS contributed to the data treatment, JAA carried out the calculations and wrote the manuscript, and JVA and MAJ helped with the interpretation of results. All the authors have revised and commented on the manuscript.

*Competing interests.* We declare that not competing interests are present

*Acknowledgements.* This research has been funded by ATMOUNT-II project [Ref. CGL2015-65627-C3-3-R (MINECO/FEDER)] and the Project [Ref. CGL2016-81828-REDT/AEI] from the Spanish Government, and by the GuMNet (Guadarrama Monitoring Network, www.ucm.es/gumnet) observational network of CEI Moncloa campus of International Excellence. Jon A. Arrillaga is supported by the Predoctoral Training Program for No-Doctor Researchers of the Department of Education, Language Policy and Culture of the Basque Government (PRE_2017_2_0069, MOD = B). We thank the contribution of all the members of the GuMNet Team, especially Dr. J.F. González-Rouco, and *Patrimonio Nacional* for the facilities given during the installation of the meteorological tower.

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

**Table 1.** Specifications about the variables measured and the devices employed in this study over the intensive summer 2017 campaign [22/06–26/09].

| Variable | Height (m, agl) | Sampling rate | Instrument | Model |
|---|---|---|---|---|
| Air temperature | 3, 6, 10 | 1 Hz | Aspirated thermometer | Young 41342 |
| Relative humidity | 2 | 1 Hz | T/RH probe | Rotronic HC2-S3 |
| Wind speed | 3, 6, 10 | 1 Hz | Cup anemometer | Vector A100LK |
| Wind direction | 10 | 1 Hz | Wind vane | Vector W200P |
| Turbulent parameters | 4, 8 | 10 Hz | IRGASON | Campbell |
| Rain | surface | - | Pluviometer | OTT Pluvio$^2$ |
| Soil moisture | -0.04 | 10 min | Reflectometer | CS655 |
| Radiation components | 1, 2 | 1 Hz | 4-component radiometer | Hukseflux NR01 |
| $CO_2$ concentration | 4, 8 | 10 Hz | IRGASON | Campbell |
| Water-vapour concentration | 4, 8 | 10 Hz | IRGASON | Campbell |

**Table 2.** Algorithm for thermally-driven downslope criteria. First column indicates the filter number; second column the physical description of the filter; and third column, the criteria to be fulfilled in order to pass each of the filters.

| Filter | Criteria | Description |
|:------:|:--------:|:-----------:|
| 1 | Weak large-scale winds | $V_{850} < 6$ m s$^{-1}$ |
| 2 | Days without synoptic cold fronts | $(\Delta\theta_{e,850}/\Delta t) > \text{-}1.5\,°$ C / 6h |
| 3 | Non-rainy events | $pp < 0.5$ mm/day |
| 4 | Minimum persistence in the downslope direction | $WD \; \epsilon \; [250°\text{–}340°]_{2h}$ |

**Table 3.** Classification of the downslope events according to their maximum 10-min averaged wind speed at 6 m from the onset until 2400 UTC.

| Type | Definition | Number of events |
|---|---|---|
| Weak | $U_{max} < 1.5$ m s$^{-1}$ | 14 |
| Moderate | $1.5$ m s$^{-1} \leq U_{max} \leq 3.5$ m s$^{-1}$ | 23 |
| Intense | $U_{max} > 3.5$ m s$^{-1}$ | 3 |

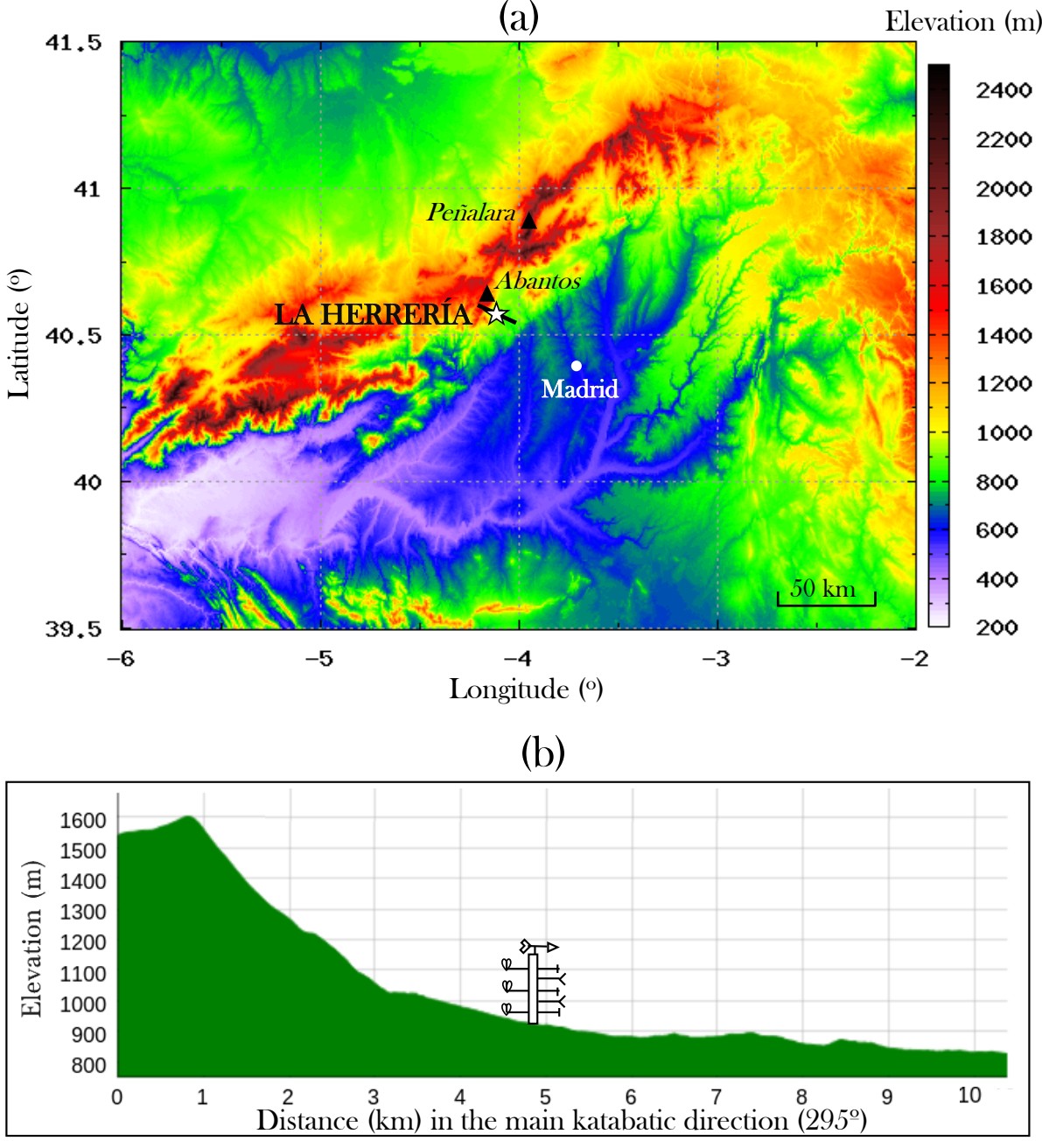

**Figure 1.** (a) Topography of the area surrounding La Herrería site, which is indicated with a star. The position of the city of Madrid, and Abantos (1763 m) and Peñalara peaks (2420 m) are additionally pointed out. The black line points the cross-section along the main downslope direction from La Herrería, represented below. The source of topography data is SRTM 90 m DEM (http://srtm.csi.cgiar.org/). (b) Topographical profile along the cross-section indicated by the black line. Obtained from *Geocontext-Profiler*.

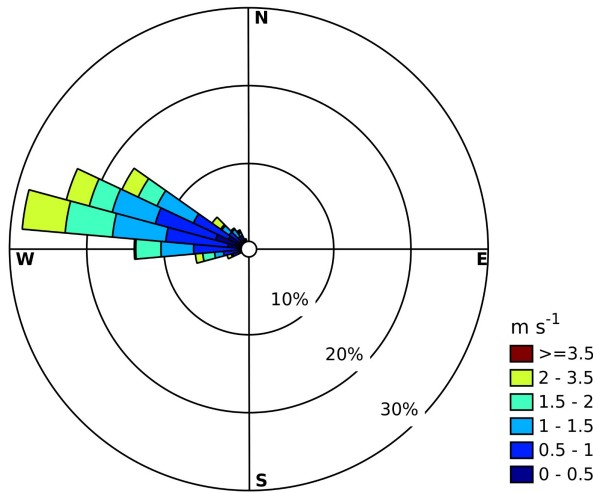

**Figure 2.** Wind rose at 6 m over the 2 h after the onset of the downslope flow for the 40 selected events.

(a) Onset

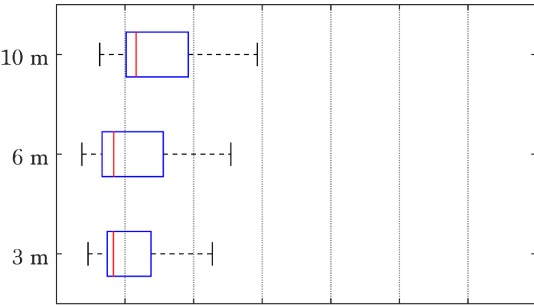

(b) Maximum intensity (considering values at 6 m).

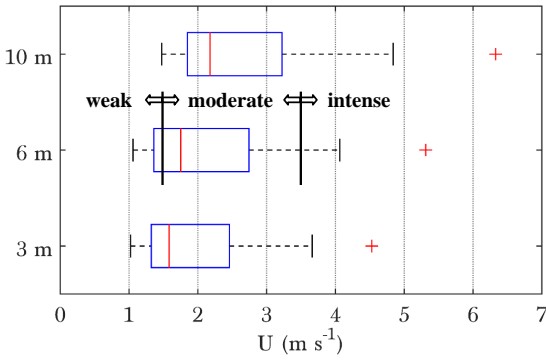

**Figure 3.** Box plots of the wind speed profile at 3, 6 and 10 m for the downslope events, (a) at the time of the onset and (b) at the time of the maximum value at 6 m (from the onset till 2400 UTC) for each event. The red vertical line within blue boxes represents the median, the blue box delimits first and third quartiles, and whiskers delimit the most extreme points not considered outliers (red crosses). Black vertical lines and arrows pinpoint the limits for the wind speed at 6 m that separate weak, moderate and intense downslope events.

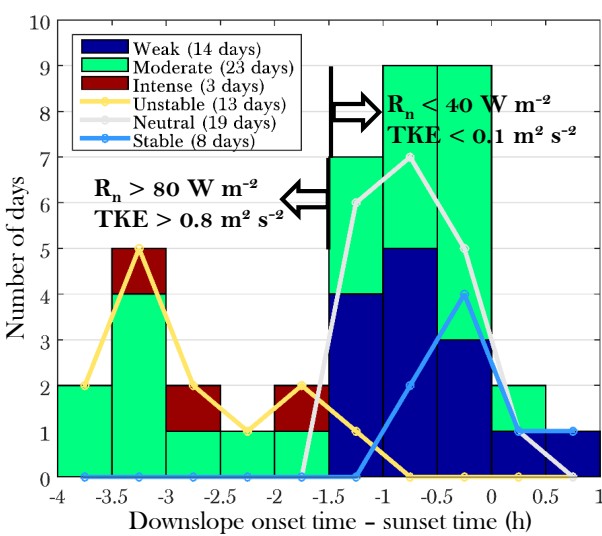

**Figure 4.** Histograms of the difference between the downslope onset time and sunset time, for the three groups of intensities (bars) and for different static stabilities at the moment of the onset (lines). Values of the net radiation ($R_n$) and turbulent kinetic energy (TKE) are also indicated on both sides from the -1.5 h time difference.

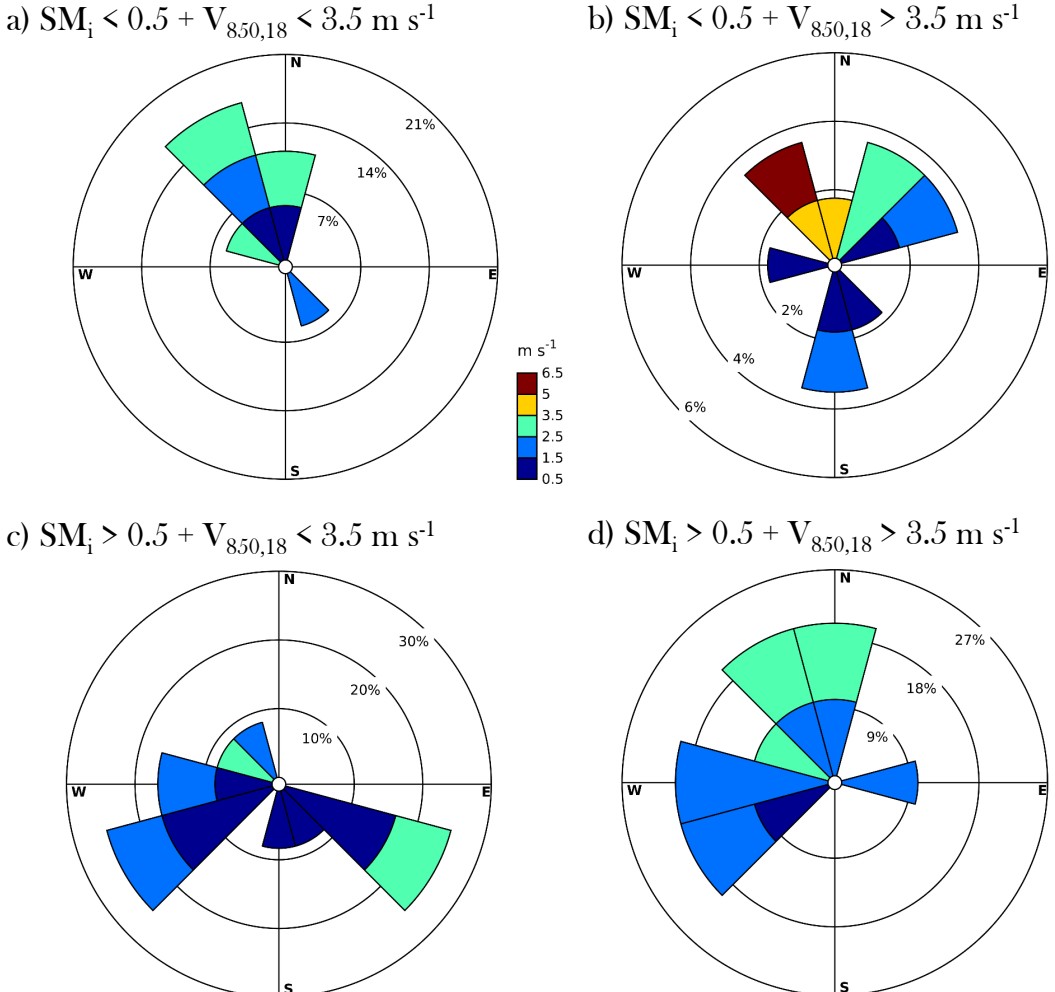

**Figure 5.** Wind roses representing the maximum downslope wind speed at 6 m in colours, for different directions from the NCEP-reanalysis wind at 850 hPa, at the closest grid point from La Herrería (40.5° N, 4° W) at 18 UTC. Wind roses are shown for different values of the soil-moisture index ($SM_i$) and the reanalysis wind speed ($V_{850,18}$) (a-d). Note that the frequency scale of the wind roses is variable.

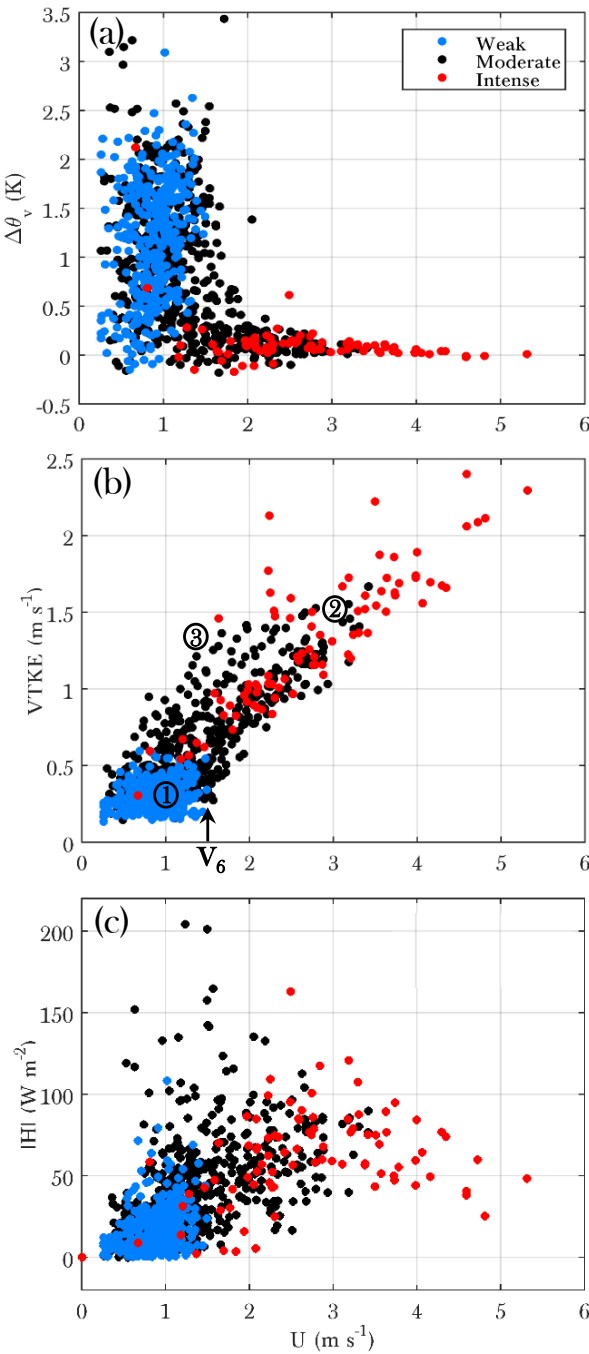

**Figure 6.** (a) Thermal stratification $\left(\Delta\theta_v = \theta_v(10m) - \theta_v(3m)\right)$, (b) turbulence velocity scale ($V_{TKE}$), and (c) the absolute value of the sensible-heat flux ($H$) at 8 m *vs* wind speed ($U$) at 6 m. The numbers in (b) pinpoint the SBL regimes defined in Sun et al. (2012): (1) weak turbulence driven by local instabilities, (2) intense turbulence driven by the bulk shear, and (3) moderate turbulence driven by top-down events. The threshold wind-speed ($V_6$) at which the HOST transition occurs is indicated too.

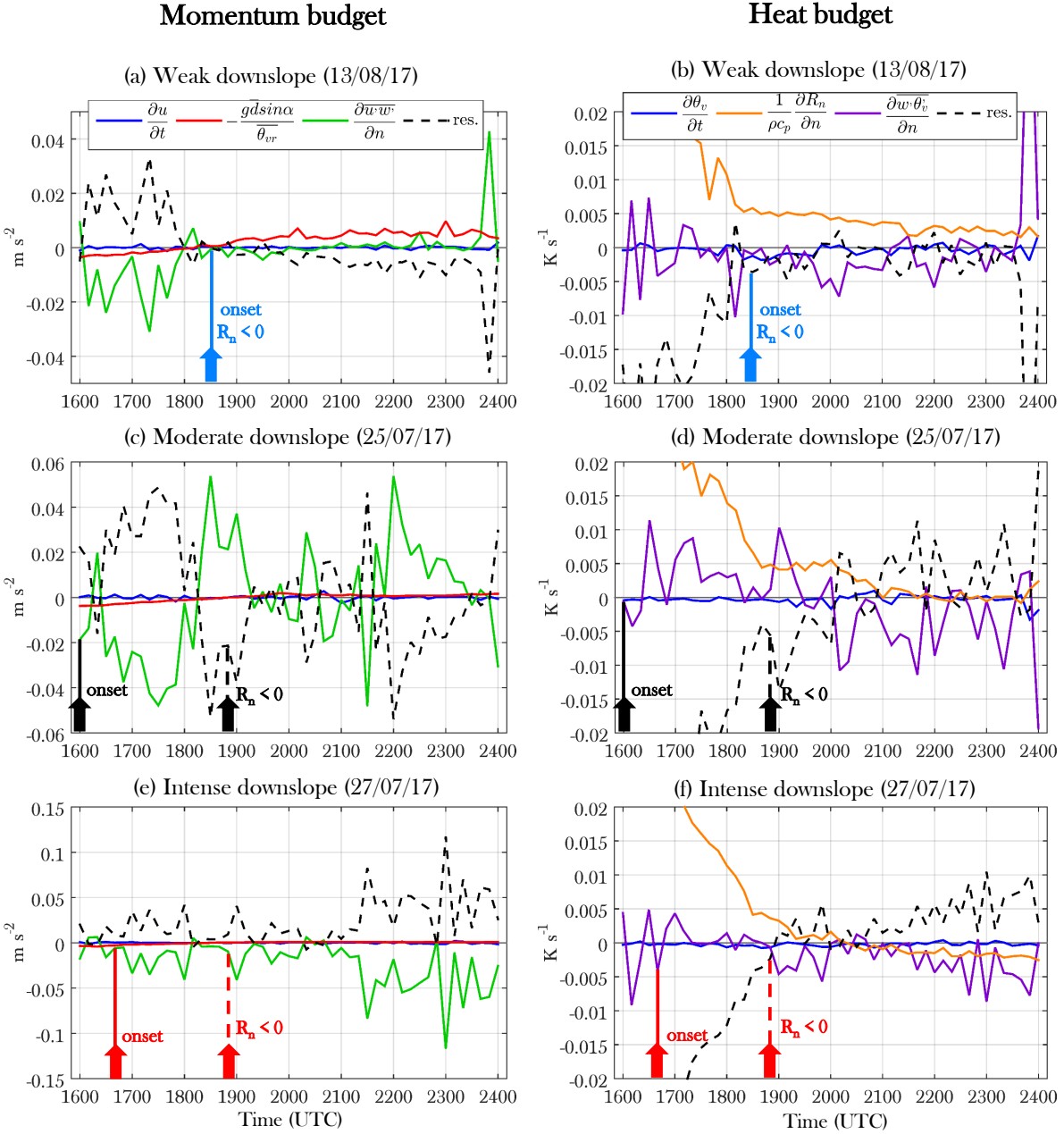

**Figure 7.** Time evolution of the different terms from the momentum (a,c,e) and heat budget (b,d,f) for the weak downslope (a,b), moderate downslope (c,d) and intense downslope (e,f) events. The solid arrows point the downslope onset time (with the vertical solid line) and the time at which $R_n$ turns negative (with the dashed vertical line).

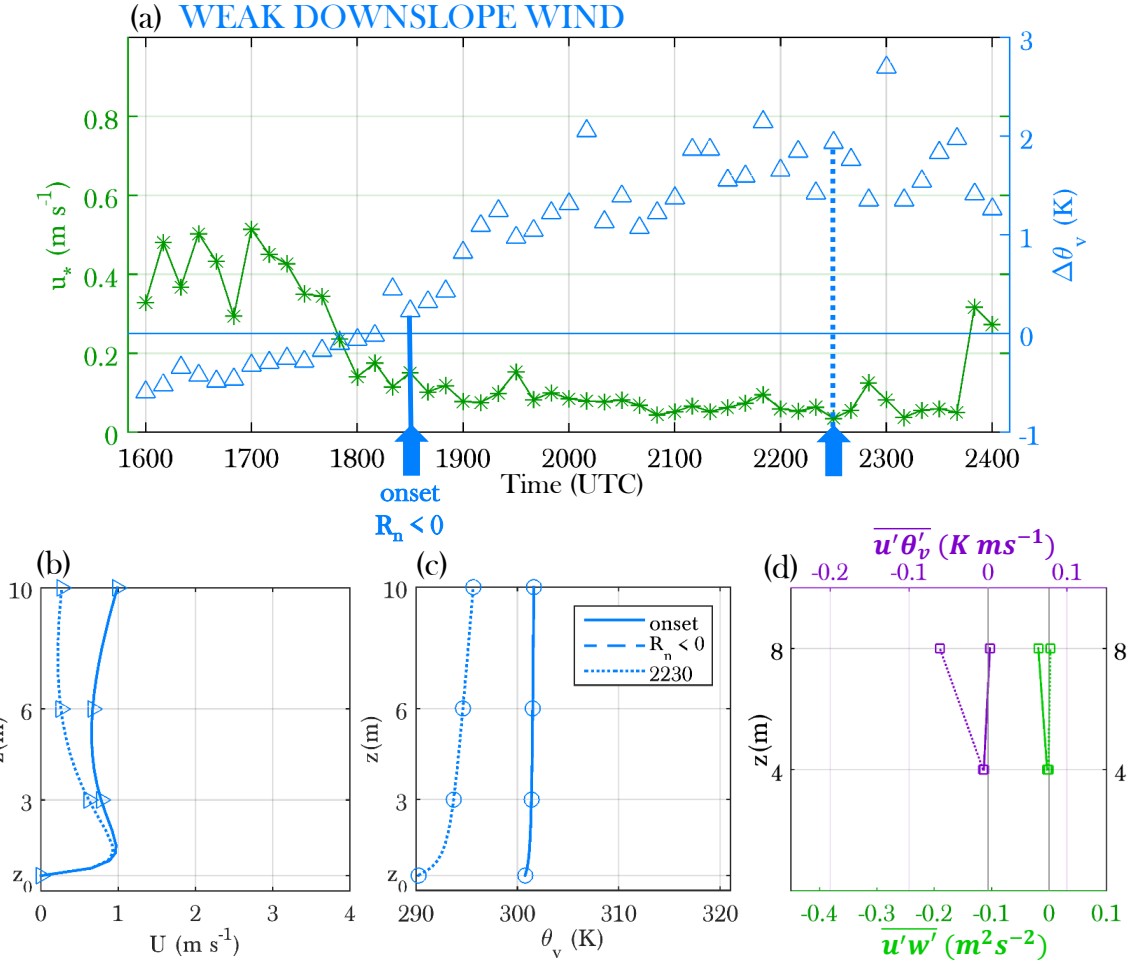

**Figure 8.** (a) Time evolution of the friction velocity ($u_*$) and thermal stratification ($\Delta\theta_v = \theta_v(10m) - \theta_v(3m)$) for the weak downslope event (13/08/2017). The horizontal line represents $\Delta\theta_v = 0$. The solid arrow with the tied vertical lines point three times of interest: the solid line indicates the onset time, the dashed line the time at which $R_n$ turns negative, and the dotted line is depicted at 2230 UTC when the SBL is already well developed. Vertical profiles of (b) the wind speed ($U$), (c) virtual potential temperature ($\theta_v$) and (d) $\overline{u'w'}$ and $\overline{u'\theta_v'}$ turbulent fluxes at the three times of interest are shown below.

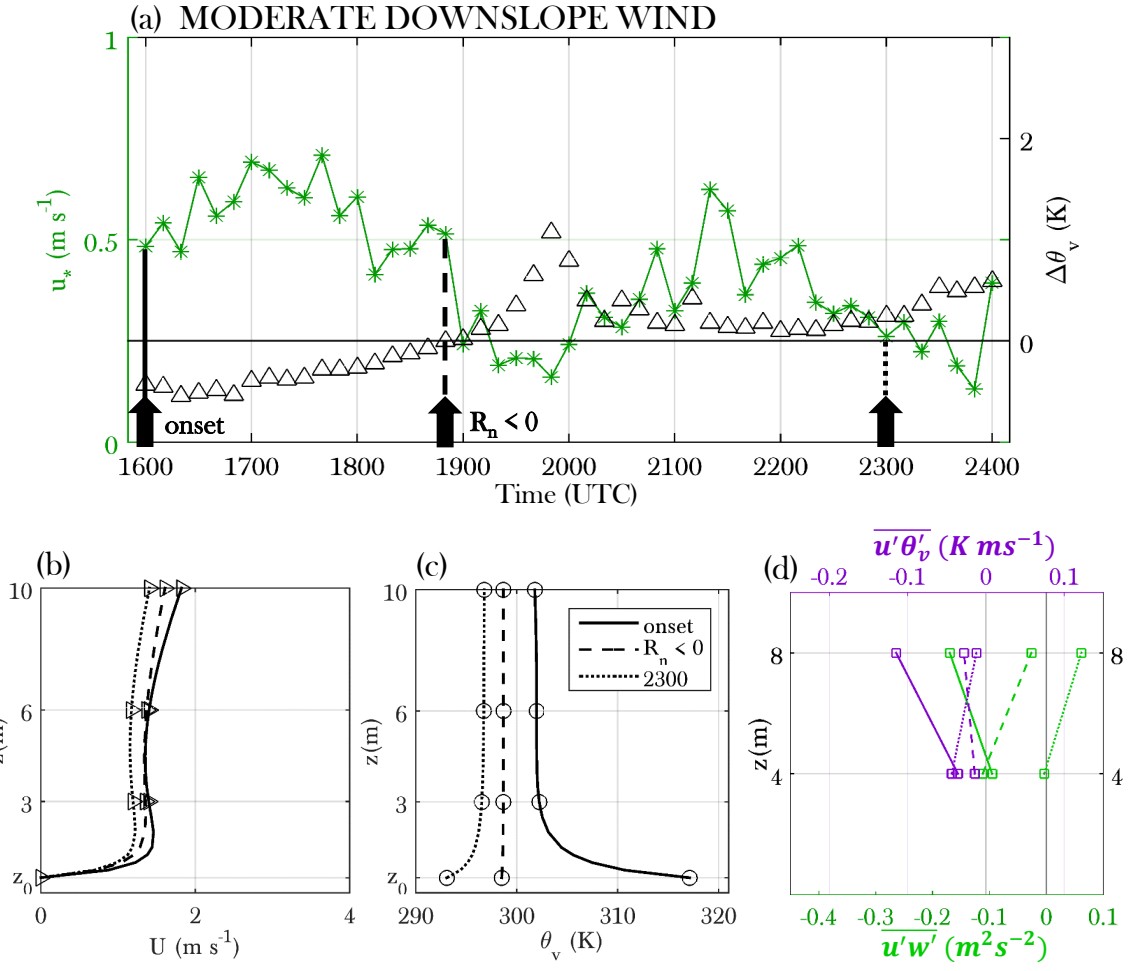

**Figure 9.** (a) *Idem* from Fig. 8 for the moderate downslope event (25/07/2017), except the change of the last time of interest from 2230 to 2300 UTC.

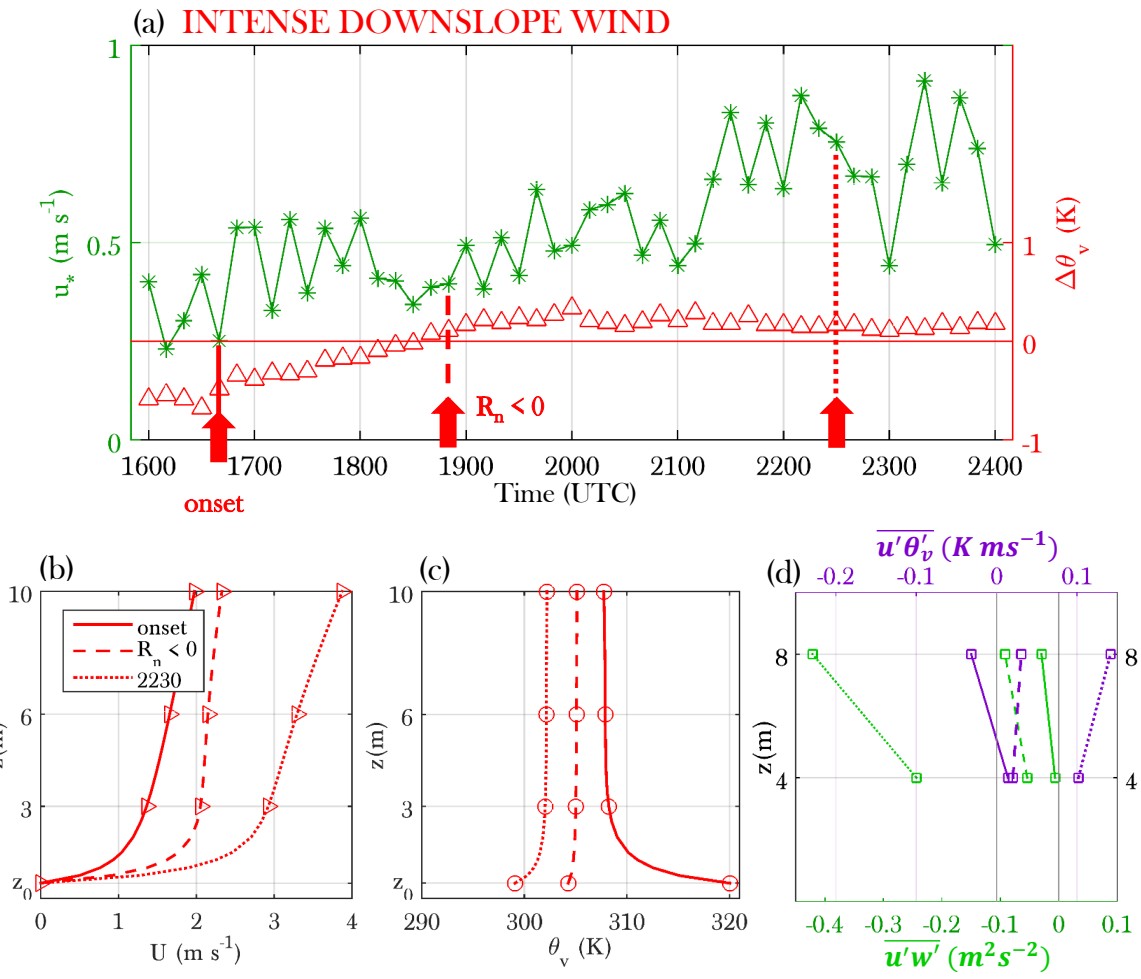

**Figure 10.** (a) *Idem* from Fig. 8 for the intense downslope event (27/07/2017).

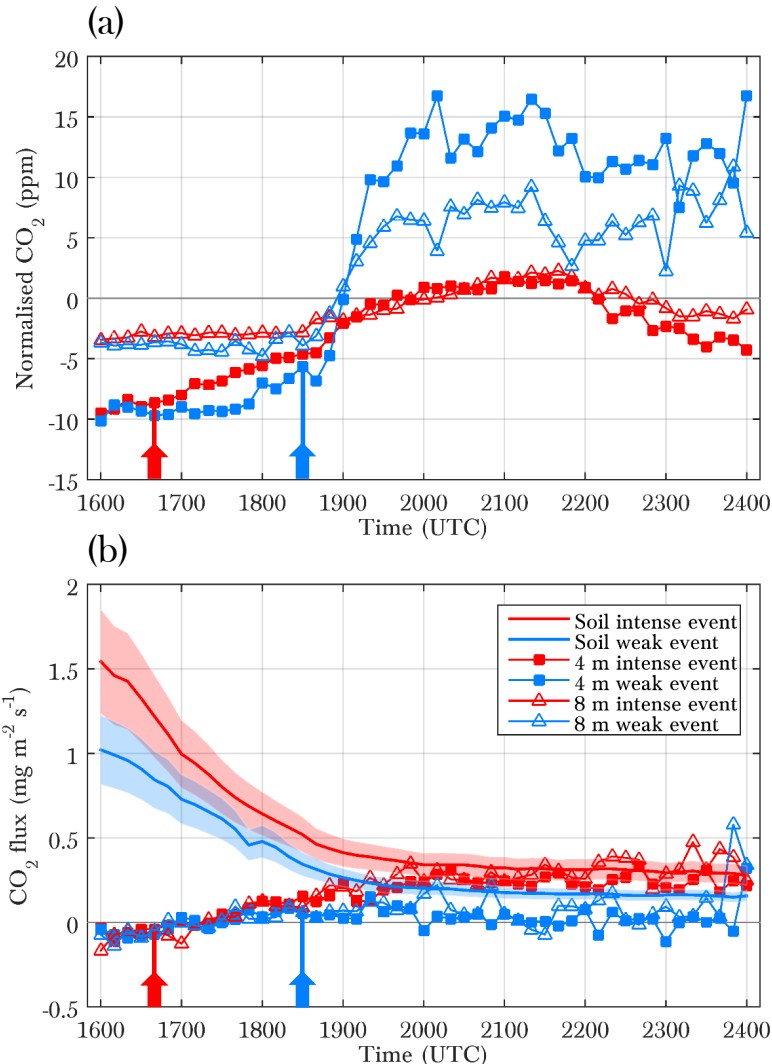

**Figure 11.** Time evolution of the normalised $CO_2$ mixing ratio (a) and vertical turbulent fluxes (b) at 4 and 8 m agl for the weak (blue) and intense (red) events. In (b) we include in solid lines the soil-respiration estimation and the 20% uncertainty of $R_{10}$ (see Eq.3) in shaded. The onset of the downslope flow is indicated from faced arrows of respective colours.

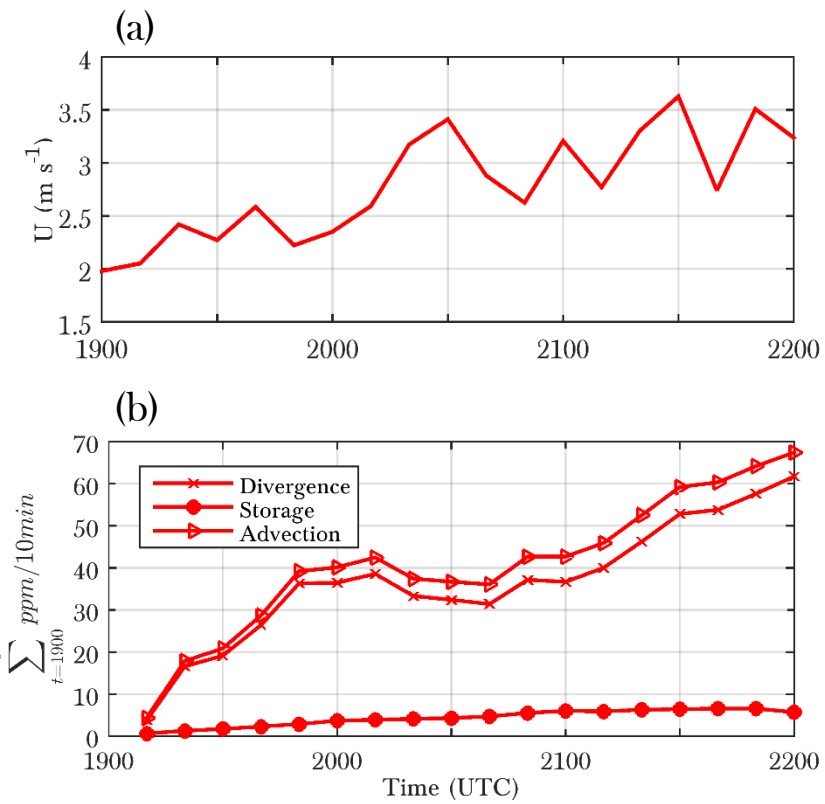

**Figure 12.** Time evolution between 1900 and 2200 UTC for the intense downslope event, of (a) the wind speed ($U$) at 6 m, and (b) the different terms of the $CO_2$ budget from Eq. 5. Note that the cumulative sum from 1900 UTC is represented in (b), and that the advection term, which is negative, is shown in absolute values.

### a) Weak downslope event

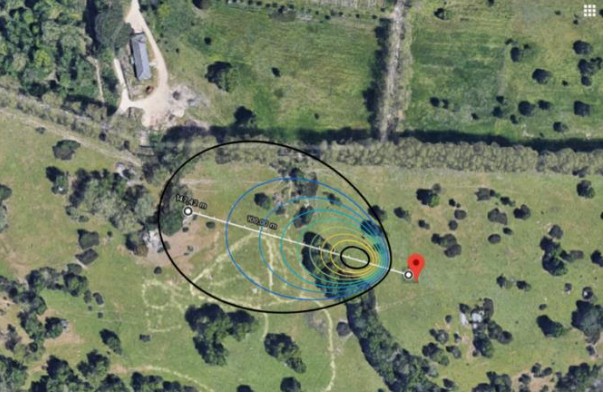

### b) Moderate downslope event

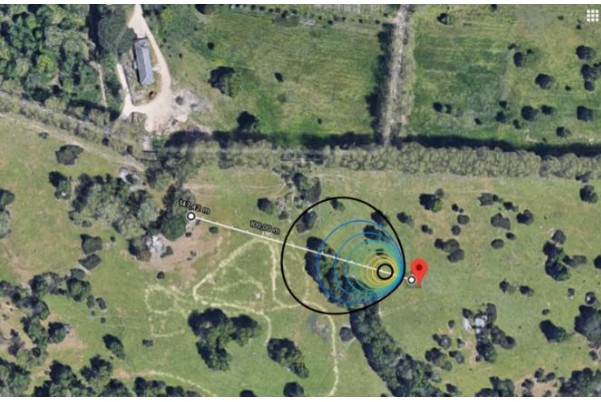

### c) Intense downslope event

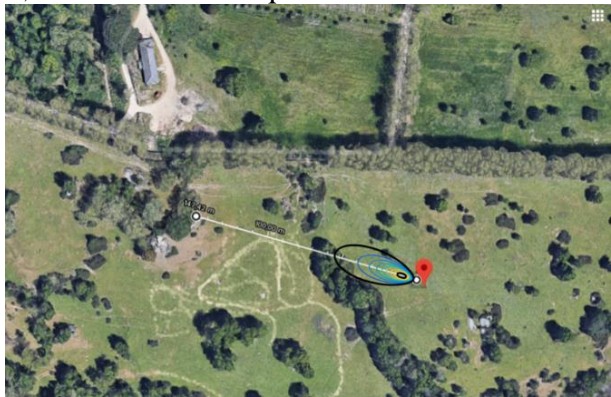

**Figure A1.** Estimation of the footprint area for the turbulent fluxes at 8 m in the mean local downslope direction (295º) in La Herrería for (a) a weak (13/08/2017), (b) a moderate (25/07/2017) and (c) an intense (27/07/2017) downslope event. Black contour lines delimit 90% and 10% of the total flux footprint. Map data ©2019 Google, Inst. Geogr. Nacional (Spain), *accessed 19 February 2019*.

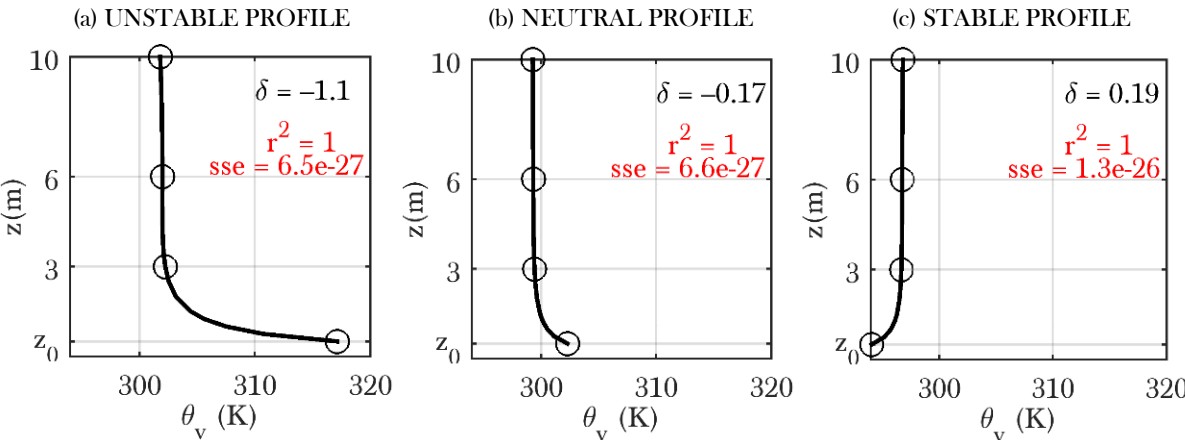

**Figure B1.** Vertical profiles of the virtual potential temperature ($\theta_v$) for different static stabilities. Values are taken from a moderate downslope event (25 July) at (a) 1600 UTC, (b) 1830 UTC and (c) 2130 UTC. The value of $\delta$ is shown in black, and the square of the multiple correlation coefficient ($r^2$) and the sum of the squares due to error (*sse*) are shown in red, as a measure of the goodness of fit.