# Peer review of "From weak to intense downslope winds: origin, interaction with boundary-layer turbulence and impact on CO2 variability"

_Atmospheric Chemistry and Physics, 2018_

## Referee Comment (RC1) · Anonymous Referee #1 · 20 Nov 2018

REVIEW

Title: Weak and intense katabatic winds: impacts on turbulent characteristics in the stable boundary layer and CO2 transport Authors: Jon A. Arrillaga, Carlos Yagüe, Carlos Román-Cascón, Mariano Sastre, Gregorio Maqueda, and Jordi Vilà-Guerau de Arellano Manuscript: acp-2018-944 Journal: Atmospheric Chemistry and Physics (ACP)

General Assessment:

The paper by Arrillaga et al. focuses on the study of the katabatic flows on a relatively flat area 2-km away from the steep slopes of the Guadarrama Mountain Range (Spain).

[Figure]

Authors discuss weak, moderate and intense katabatic events. The manuscript also examines a horizontal CO2 transport driven by the katabatic advection.

Judgement:

I think that obtained results may be useful for further understanding of the katabatic flows and manuscript is suitable for publication in the ACP, however not before a major revision. I recommend acceptance with major revisions although most of the comments are not that major and related to lack of clarity. My specific comments are listed below.

Revision issues:

Although it is appropriate to refer readers to other papers for the details of the field campaign and instrumentations, more info is needed in this paper than is currently provided (see my remarks detailed below).

Data (post) processing. Data processing is only briefly mentioned (p. 5). More info is needed in this paper for the details of the turbulent flux measurements than is currently provided. A reader (in order to acknowledge the results) would want to know: 1) how the filtering was done (block average, high-pass, other?), 2) what data-quality control checks were used, 3) how the wind stress (or momentum flux, friction velocity) was computed in Fig. 11? Based only on the longitudinal (or downstream)  wind stress component or both longitudinal and lateral (or crosswind) <v'w'> stress components? Why?

Large errors in the measurement of the turbulent fluxes can result from relatively small errors in the alignment of a sonic anemometer due to the cross contamination of velocities (i.e. fluctuations in the longitudinal wind speed components appear as vertical velocity fluctuations, and vice versa). To avoid these errors rotation of the anemometers' axes is needed to place the measured wind components in a streamline coordinate system. The most common method is a double rotation of the anemometer coordinate system to compute the longitudinal, lateral, and vertical velocity components (Kaimal

and Finnigan, 1994, section 6.6). Was this done?

Since the sonic anemometer measures the so-called 'sonic' virtual temperature (which is close to the virtual temperature) the moisture correction in the sonic anemometer signal is necessary to obtain the correct value of temperature itself and sensible heat flux (e.g. Kaimal and Finnigan, 1994). Authors reported the sensible heat flux (Figure 9). To value the present results the authors should either show that the moisture corrections and their impact on the results are small, or (if otherwise) apply moisture corrections to the sonic temperature following Schotanus et al. (1983) based on the data collected by the Campbell fast-response open path infrared gas analyzer listed in Table 1.

Authors say nothing about the Webb correction (also referred as WPL or Webb effect after the paper by Webb et al. [1980]). This correction must be taken into account when the turbulent fluxes of minor constituents such as carbon dioxide or, in some cases, water vapor are measured (Webb et al. 1980).

In a slope-following coordinate system, the horizontal (along the slope) heat (buoyancy) flux contributes to the net buoyancy term and, therefore, the Monin-Obukhov stability parameter $z/L$ (see page 11 and Fig.10) contains this additional term (e.g., see Grachev et al. 2016, their Eq. (3) and references therein). Authors say nothing about this issue for calculation $z/L$ which is very important point for katabatic flows.

Minor and editorial/technical comments:

Page 5, Line 8. Replace $CO^2$ by $CO_2$.

Page 11, Line 28. I suggest to provide a definition of the Monin-Obukhov stability parameter ($z/L$) and the bulk Richardson number ($R_B$) for a layman reader.

Page 12, Lines 2-9. I would like also to see here a discussion on difficulties and controversy of interpretation associated with the critical Richardson number (e.g., Grachev et al. 2013 and references therein).

[Figure]

References: Replace 'Boundary Layer Meteorol.' by 'Boundary-Layer Meteorol.' (dash is missed).

Page 19, Line 1. Please correct Silvana's name: Di Sabatino S. instead Sabatino, S. D.

Page 20, Line 9. Please correct reference Pardyjak et al. as follows: Pardyjak E.R., Fernando H.J.S., Hunt J.C.R, Grachev A.A., Anderson J.A. (2009) A case study of the development of nocturnal slope flows in a wide open valley and associated air quality implications. Meteorologische Zeitschrift, 18(1), 85–100. DOI: 10.1127/0941-2948/2009/362

Included references:

Grachev A.A., Andreas E.L, Fairall C.W., Guest P.S., Persson P.O.G. (2013) The critical Richardson number and limits of applicability of local similarity theory in the stable boundary layer. Boundary-Layer Meteorol. 147(1): 51–82. DOI: 10.1007/s10546-012-9771-0

Kaimal J.C., Finnigan J.J. (1994) Atmospheric Boundary Layer Flows: Their Structure and Measurements. Oxford University Press, New York and Oxford, 289 pp

Schotanus P., Nieuwstadt F.T.M., De Bruin H.A.R. (1983) Temperature measurement with a sonic anemometer and its application to heat and moisture fluxes. Boundary-Layer Meteorol. 26(1): 81–93. DOI: 10.1007/BF00164332

Webb E.K., Pearman G.I., Leuning R. (1980) Correction of flux measurements for density effects due to heat and water vapour transfer. Q. J. R. Meteorol. Soc. 106(447): 85–100. DOI: 10.1002/qj.49710644707
* * *

---

## Referee Comment (RC2) · Anonymous Referee #2 · 29 Dec 2018

Review of the "Weak and intense katabatic winds: impacts on turbulent characteristics in the stable boundary layer and CO2 transport" by Jon A. Arrillaga, Carlos Yagüe, Carlos Román-Cascón, Mariano Sastre, Gregorio Maqueda, and Jordi Vilà-Guerau de Arellano

Manuscript number: acp-2018-944

General Comments The authors study downslope flows at a site close to the Guadarrama Mountain Range by means of mean and turbulence measurements at a 10 m tower. The authors use an algorithm to identify periods with low synoptic forcing and katabatic flows, and then separate the periods into those with weak, intermediate and

strong katabatic flows. Finally, they study the conditions under which each of these occurs and the associated boundary layer structure and CO2. The study shows some interesting, though not surprising results on the correlation between the strong wind episodes that lead to weakly stable stratification, and weak wind episodes that lead to very stable stratification. Still, the study has major gaps and in general lacks a thorough analysis of physical processes and associated budgets needed to substantiate the explanations which are at the moment sometimes given without a thorough proof (see specific comments below on whether this flows are katabatic at all and what their origin is given that katabatic flows cannot develop in unstable stratification, on the need to look at budgets, or for example the text connected with Figure 3). Also, there is a lack of thorough understanding of the nature of these flows (both in terms of the driving mechanisms and the interaction with turbulence) and the fact that the location of the jet maximum is a vital information that is lacking from the entire study, if indeed this flows are shown to be katabatic. If the turbulence data are collected from above the jet maximum, then it is turbulence that is not connected with the ground and therefore is not expected to show standard boundary layer characteristics (such as MOST etc). Where the jet maximum within the tower depth for each averaging period is needs to be added in the discussion of all the results and all the discussions and conclusions adjusted accordingly. For a more in depth study of the turbulence characteristics of katabatic flows Grachev et al. (2016) paper gives excellent information.

Specific Major Comments 1. Turbulence data processing As already mentioned by the first reviewer, the authors fail to give vital information on the turbulence data processing. Apart from the missing information on rotation methods and turbulence corrections, the authors also fail to motivate why they use a 10min averaging time, which in stable boundary layer is generally too long, and even for strong wind conditions the more appropriate averaging time would be 5 min, while for weak winds it is most likely 1 min or less. I suggest the authors calculate ogives or multi-resolution flux decomposition (e.g., Vecenaj et al, 2012) for their 40 episodes and estimate the most appropriate averaging time. If they need to have 10 min averages for comparison to the slow sen-

sors, then the fluxes can be afterwards averaged to that value. 2. Footprint analysis 1. The authors themselves talk about the footprints influencing the values of the fluxes (Pg 4, ln 33 or Pg. 14, ln 23-24), however, no footprint analysis is provided. I suggest the authors use the footprint model of Kljun et al. (2015) to examine the differences in the source area of the turbulent fluxes for the three different categories. But on another note, the sentence on Pg. 4 is erroneous: the footprint does not induce "uncertainty in estimation of fluxes" if the fluxes are calculated from the eddy covariance, they might just represent turbulence originating from other locations. 3. Structure of the katabatic flow I find myself wondering if not doubting if the flow the authors are studying should be classified as katabatic at all or not. Katabatic flows possess very specific characteristics: a low level jet, formation due to surface temperature deficit and retardation due to surface friction (turbulent momentum transport towards the surface), very specific turbulent structure associated with its jet profile: negative momentum and positive horizontal heat flux below the jet and the opposite above, minimum in TKE at the jet maximum (see Grachev et al. 2016). The profiles in Fig. 11, particularly for the strong cases do not resemble katabatic ones at all, and the weak cases have a low level maximum that could be also just due to the interpolation scheme, and then appears to have a secondary maximum above the height of the tower. On a side note: why are the sonic measurements not used in the wind speed profiles such as in Figure 11? Could it be that it actually is the basin flow (Pg 2, ln 15. How do you ascertain that your flow is not actually influenced by the Madrid basin and is purely katabatic). The authors should look at the profiles of the turbulence quantities to first identify if their flow qualifies as katabatic and second to actually show if their profiles in Fig. 11 are physical at all or the low level jet in the weak case is purely a construct of interpolation scheme, and what is happening with the secondary maximum. The profiles of turbulent quantities would also allow them to estimate the jet maximum height in each individual period. The jet maximum height is indeed the vital parameter when studying anything related to katabatic flows since at jet maximum wind speed will be maximal but the turbulence will be zero – and thus exactly the opposite of standard flatterrain stable boundary layer structure that the authors so heavily rely on in the HOST, MOST, shear capacity and other diagnostics. In that respect, if there is really a low level jet maximum below 3 m in the weak cases, then the turbulence above that height might be disassociated with the surface and therefore not exhibit standard boundary layer characteristics. Not taking this fact into account invalidates the conclusions. 4. Study of budgets The authors should present budget of the momentum and heat to substantiate their claims (such as the section 3 and 4 when talking about the development of the flow and its interaction with turbulence and transition to very or weakly stable boundary layer), and also to more fully understand the processes at hand. By examining the budgets of the katabatic flow one could isolate the importance of individual terms (local generation, dissipation, advection etc) on the weak, intermediate and strong katabatic flows and therefore show if the weak katabatic flow for example is locally driven and the strong katabatic flow is advected from the steep slopes 2km away, whereby the change in slope (from 25° to 2°) leads to the deepening of the flow as observed by Smith and Skyllingstaad (2005). The budgets will also show the importance of mesoscale and not just the large-scale pressure gradients on the flow, even if only as a residual term. The budgets could answer where the claims that stronger unstable turbulence facilitates intense katabatics. This indeed is counterintuitive as for the katabatic flow to develop one needs a large temperature deficit (i.e. cooling) and turbulence suppresses the katabatic flow while unstable stratification does not even allow the development of katabatic flow (Pg. 8, ln 7-13). 5. Stratification How was the virtual potential temperature calculated? Was the humidity needed to convert air temperature to virtual potential temperature used from Irgason and at which level? Also about the calculation of the potential temperature gradient: On Pg. 7, ln 20 and Equation 1, if you are using a 3th order polynomial why is the stratification calculated only from delta? The true temperature gradient dTheta/dz (if one takes the derivative of Eq 1) has contributions from beta, gamma and delta and depends on height. 6. Origins of the flow Tied to the previous comments, the paper fails to determine the origin of the katabatic flows. For example, on Pg. 8, ln 5 the authors mention that the stratification is unstable

and net radiation positive during the onset of the strong katabatic episodes. Given that the katabatic flow is caused by stable stratification (temperature deficit) and therefore cannot develop in unstable stratification, the authors should show evidence of why they think their flow is katabatic, and how and where it originates from (does it originate on the steep slope where the stratification has already turned stable due to shadowing? And is now merely advected to the study site?) 7. Figure 7 and the correlation to soil moisture Putting the 4th order fit through the data presented in Figure 7 is stretching it beyond any justifiability. Indeed, the spread of the data is so large that a linear fit would be possible at maximum to show that for low soil moisture G is slightly positive, but for high it is mostly negative. The results for longwave-radiative loss show no correlation between the data, both linear or non-linear. I therefore protest against any conclusions based on these fits and the identified two maxima. 8. Energy balance closure and the ground heat flux It is interesting to note that the level of energy balance closure is so high in the study site. How did the authors calculate the ground heat flux and the heat storage in the layer above the heat flux plate? Given the nature of the weak and the strong flows, one could also argue on the importance of advective processes, but the energy balance appears to suggest that advection is not important for strong katabatic flows. 9. Interaction with turbulence The second motivation of the study talks about the interaction between turbulence in SBL and katabatics but actually, the relevant turbulence that is interacting was found to be unstable stratification before the onset of the wind itself. 10. Language I suggest a professional or native speaker to check the language as I didn't want to enumerate all the things that need to be changed (e.g. "avoids" should be "prevents", "striked" should be "stricken" or rather something more appropriate, "fogs" as plural does not exist, it is always "fog", "emplacement" sounds awkward etc.)

Individual Major Corrections 2. Pg. 1, ln 1. What are the "dynamic and turbulent" features of SBL? 3. Pg. 1, ln 8: In Figure 6, the limits is 3.5 m/s not 6. What is the correct number? 4. Pg. 3, ln 8-14: No, even over flat terrain with the existence of a low level jet or even in very stable stratification without a low level jet MOST is

not valid. 5. Pg 4, ln 24: why would a strong surface thermal inversion necessarily be allowed to develop just because the slope angle is low, if there is enough wind to prevent its development? 6. Pg. 4, ln 35: you have not mentioned the CO2 fluxes at all until the moment when you mention the negligible effect of the urban area. Or do you mean the effect of urban area on all the fluxes? 7. Figure 3. Why is there no data from the sonics? Also, the way the data are presented all lumped together does not show if there are wind maxima at different heights for the different episodes, periods. I suggest the authors calculate the jet maximum if possible, normalize the height with it and then plot the normalized profiles. 8. Pg. 6, ln 20-26: Nowhere in Figure 3 is it visible that there is a development of skin flow or where the jet maximum is. Indeed, the figures seem to suggest that the jet maximum is always above 10m. 9. Figure 3 should also include the information on the temperature profiles and turbulence. 10. Pg 7, ln 6-7: I find it very strange that the results at 3 and 10 m do not show conformation to the classification and only 6m is so good. The sonic data should be used to study this more in detail and give a physical explanation why this is so. 11. Flocas et al. reference is missing from the list of references 12. Pg. 7, ln 17: is it the soil or the skin temperature? 13. Pg. 8, ln 1: is the low value of TKE due to the jet maximum being close to the measurement height or because one is above the jet? 14. Pg 8, ln 6: katabatic flow develops turbulence through shear generation. What you mean to say by "relation between katabatic flow and turbulence" I guess is, the turbulence before the onset of the flow 15. Pg. 9, ln 3: why would large soil moisture after precipitation during nighttime lead to the enhanced cooling of the soil? 16. Pg 9, 7-13: The study on the influence of stratification needs to be associated with momentum and heat budgets of the flow to show whether the conclusions drawn are substantiated in the text. 17. Pg. 10, ln 8-19: the transition will depend on the location of the jet maximum and if it is below 6m or above or is moving between. The exact value that is lower than in Sun is indeed no wonder given the fact that there is a jet maximum present and not in Cases-99. 18. Pg. 11, ln 1-3. The two sentences on MOST cannot be applied to the current study without more understanding of the processes studied. MOST will

only be applicable very close to the surface if the stratification is weakly stable and turbulence is well developed, and if the terrain is horizontally homogeneous. If there is a low level jet close to the tower height or even worse, within the tower height, MOST by definition is not valid as there is another height scale that is more important than $z/L$. 19. Pg. 11, ln 25: Isn't the fact that Fig 10b resembles Fig. 8b by construction since you change the definition of shear capacity to match the HOST? 20. Pg 12, ln 4-6: The calculation on Rb will again depend on the existence of the jet maximum if it is below, and therefore it doesn't make much sense as a measure. A better measure would be the gradient Richardson number Ri which the authors could calculate from the interpolated profiles and therefore obtain a profile of Ri. 21. Pg 12, ln 24: say that the value of the diurnal peak is not shown, or do you refer to the little part before the transition shown in Fig. 11? 22. Pg. 12, ln 25: does the flow arrive or develop? Show the budgets 23. Pg. 12, ln 1-2: how accurate is this wind maximum that does not exist in the measurements but only in the interpolation? 24. Pg. 12, 5-6: In Grachev et al. (2016) paper it says that the flow is stationary but only in the well-developed phase. You are focusing on the transition. 25. Pg. 15, ln 23: "form when maximum wind speed is kept" should be "have maximum wind speeds below", because indeed you are talking about the katabatic wind speed of 1.5 and not the ambient wind speed into which the katabatic wind impinges. 26. Pg. 15, 26: Wind shear is not driving katabatic flow, the driver is the negative buoyancy and wind shear is the product of the katabatic flow itself 27. Pg. 15, ln 29: "intense katabatics are found" should be "int. kat. have maximum wind speeds..". It is again a question of cause and effect 28. Pg. 16, ln 10: the scaling regime expected to be valid for at least very stable conditions is local or most likely z-less scaling. 29. Pg. 16, ln 25: influence of submesoscale phenomena will be visible in the calculated ogives or multi-resolution flux decomposition.

Extra references: Kljun, N., Calanca, P., Rotach, M. W., and Schmid, H. P.: A simple two-dimensional parameterisation for Flux Footprint Prediction (FFP), Geosci. Model Dev., 8, 3695-3713, 2015. Večenaj, Ž., Belušić, D., Grubišić, V. et al. Boundary-Layer Meteorol (2012) 143: 527. https://doi.org/10.1007/s10546-012-9697-6

---

## Referee Comment (RC3) · Anonymous Referee #3 · 17 Jan 2019

General Comments: The manuscript investigates katabatic flows on the basis of 40 occurrences observed during one summer season at the foothills of the Guadarrama Mountain Range in Spain. The data set has been split up into weak, moderate and intense events, based on the observed maximum wind speed observed during each individual case and is then analyzed under various aspects. The study shows distinct differences between the different classes of katabatic flow, the number of intensive katabatic flow cases is, however, very low (3) and rises thus questions on the statistical significance of the reported results. The paper is in general well structured and includes a good literature overview on the subject. The figure layout is in general rather inhomogeneous over the paper and should be reworked. Several of the figures are in

addition hard to read, mainly due to small labels and legends. Finally, the manuscript requires a thorough makeover by a native English speaker. Main points in this context are rather complicated sentence structures, unconventional wording obviously taken from the dictionary (e.g. emplacement instead of site/location), missing commas, the improper use of prepositions and articles, and grammatically incomplete sentences. I have marked a quite a few, but far from all, instances in my specific comments.

Specific comments:

1) P1, L1: "on the dynamics" instead of ""in the dynamics"

2) P1, L5: insert comma after "moderate and intense"

3) P1, L9: insert "flow" after "katabatic"

4) P2, L26: "at contrasting" instead of "in contrasting"

5) P2, L28: insert comma after "model"

6) P2, L33: "In contrary" instead of "At contrary"

7) P3, L8 and 11: "emplacement" is rather uncommon, better "site or location"?

8) P3, L15-16: "on the concentration" instead of "in the concentration"

9) P3, L16: remove "the" before "CO2

10) P3, L16: "in coastal areas" instead of "at local areas"

11) P3, L27: sentence incomplete; "the role of. . . . . ., in CO2 mixing ratios" ; should be "in controlling/affecting CO2 mixing ratios"

12) P3, L34: insert "concentrations" after "CO2"

13) P4, L23: "a relatively" instead of "an relatively"

14) P4, L23-24: "immediately besides" is quite strange; better "close to"?

15) P4, L29-30: "needleleaved evergreen tree cover", sounds complicated; isn't it just "coniferous"?

16) P4, L34: replace "inexistent" by "absence of"

17) P5, L8: formatting error "CO2" subscript

18) P6, L5: insert "concentrations" after "CO2"

19) P6, L7: "Forty were selected as days……….." The sentence is grammatically poorly formulated and hard to read. Please rephrase

20) P7, L2: insert "the criteria" after "meet"

21) P7, L2 (and other instances): replace "minutal" by "minute"

22) P7, L4: replace "weak" by "low"

23) P7, L29: there is no Figure 5a)

24) P8, L34: Why are you presenting a 4th-order polynomial fit; any physical reasoning for this? A simple trend could also be seen from a linear regression, and the two peaks resulting from your fit seem to be rather arbitrary; Thus I see a big danger of an over-interpretation of non-existing features in the corresponding paragraphs on page 9, L1-13 (see also my comment on Figure 7)

25) P9, L17-18: "…., the shear associated with the katabatic flow increases, and the downslope flow strengthens progressively." I do not understand the direct link between this two statements; How can increase in shear strengthen the downslope flow? Might also be a misunderstanding from my side, but then the sentence should be rephrased

26) P9, L20: insert "the" before "surface"

27) You should define the calculation of VTKE already here, and not two lines under

28) P10, L2: insert comma after the parenthesis with the wind speed

29) P10, L10: "increases linearly"; I could also see a square root dependency here

30) P10, L29: replace "on" by "for" or "during"

31) P10, L32: insert comma after "night"

32) P11, L17: Sentence has to start with an upper case letter; "Van Hooijdonk….."

33) P11, L25-26: "intense and weak katabatics cluster into two clearly distinct regimes"; at 8 m I still see a considerable amount of black and red data points for SC<3 with distinct elevated VTKE levels; can you explain/comment on this

34) P12, L4: put "by definition" between commas

35) P12, L17: replace "related with" by "related to"

36) P13, L7-8: start the sentence with "In contrary, U remains….."

37) P13, L12: insert comma after "takes place"

38) P13, L25: replace "so doing" by "doing so"

39) P14, L9: insert comma after "SBL"

40) P14, L20: insert comma after "nearly 0"

41) P14, L21: insert comma after "assumption"

42) P14, L22: insert comma after the reference

43) P14, L25: insert comma after "equation"

44) P 15, L8-10: this sentence has to be rephrased, maybe even better split in to or three! In particular complicated is the part "…by the presence upwind of a land use component of forest…….."

45) Some small inconsistencies in the references

a. Boundary Layer Meteorol. vs Boundary-Layer Meteorol.; I believe the latter one is

the usual b. A few journal abbreviations are not terminated by a period; e.g. Borge et al. and Plaza et al.

c. Presentation of doi or not for articles

46) Page 22, Table 1; inaccurate caption, I suggest: "Specification of the values measured and the devices . . .. . ."; the specification of a value is not technical!

47) Page 24, Table 3; add number of occurrences for each class in the table; maybe also an idea to place the measurement frequency directly in the table for each sensor instead of using the footnote solution

48) Page 25, Figure 1a; the location names are difficult to read; use larger fonts and bold style; in addition have the degree symbols in the axes labels an underscore that should be removed

49) Page 27, Figure 3: I suggest to split this figure in 3 separate ones for the weak, moderate and intensive cases; as it is presented now you loose a lot of information by the averaging; I would also like to see the 40 individual profiles in this plots, e.g. as grey lines in the background

50) Page 31, Figure 7: I cannot see that the applied 4th-order fitting makes any sense; do you have any physical reasoning for your choice

51) Page 34, Figure 10: labels/legends too small

52) Page 35, Figure 11: a) use different line styles for 4 and 8 m (in particular important for the intense event in red); why have you selected 21:00 as last time you present; from the time series it looks like that is more a transition phase, while e.g. 22:00 appears to be a more stationary situation; b), d), f): I suggest to use a common x-axis span at least for the wind speed

53) Page 36, Figure 12: axis labels too small!

54) Page 37, Figure 13: use different line styles for 4m and 8 m

---

## Author Comment (AC1) · 21 Mar 2019

**Responses to comments from Referee #1**

**MANUSCRIPT: acp-2018-944**
**TITLE:** From weak to intense downslope winds: origin, interaction
with boundary-layer turbulence and impact on $CO_2$ variability
**AUTHORS:** Jon A. Arrillaga, Carlos Yagüe, Carlos Román-Cascón,
Mariano Sastre, Maria Antonia Jiménez, Gregorio Maqueda
and Jordi Vilà-Guerau de Arellano
* * *
**MAIN CHANGES IN THE MANUSCRIPT:**

o  Title.

o  Abstract and motivating aspects.

o  Denomination: katabatic $\rightarrow$ downslope.

o  Further information about data postprocessing in Sect. 2.2.

o  New Sect. 4 in the revised manuscript: analysis of the heat and momentum budgets, profiles and the estimation of the jet-maximum height for three representative events.

o  Summary and conclusions.

o  Appendix A (footprint estimation) and B (assessment of the thermal profile).

o  Removed figures (numbers from the old manuscript): Fig. 7, Fig. 10, Fig. 11 and Fig. 12.

o  Merged figures (numbers from the old manuscript): Figs. 4 and 5, Figs. 8 and 9.

o  New figures (numbers in the revised manuscript): Fig. 7, Fig. 8, Fig. 9, Fig. 10, Fig. A1 and Fig. B1.

o  Slightly modified figures (numbers in the revised manuscript): Fig. 1 and Fig. 11.

o  Wording and English review.

**Judgement:**

**I think that obtained results may be useful for further understanding of the katabatic flows and manuscript is suitable for publication in the ACP, however not before a major revision. I recommend acceptance with major revisions although most of the comments are not that major and related to lack of clarity. My specific comments are listed below.**

We thank Referee #1 for his/her review about the manuscript and for highlighting its suitability for publication in ACP. Responses to the specific comments are given point-by-point below, and the changes undertaken in the manuscript can be checked up both from the revised manuscript and the tracked-changes version of the manuscript provided.

**Revision issues:**

**1. Although it is appropriate to refer readers to other papers for the details of the field campaign and instrumentations, more info is needed in this paper than is currently provided (see my remarks detailed below).**

As suggested by this referee, we have included further information in the new manuscript regarding instrumentation, data post-processing and corrections. Specifications about each of the posed queries are provided below.

**2. Data (post) processing. Data processing is only briefly mentioned (p. 5). More info is needed in this paper for the details of the turbulent flux measurements than is currently provided. A reader (in order to acknowledge the results) would want to know: 1) how the filtering was done (block average, high-pass, other?), 2) what data-quality control checks were used, 3) how the wind stress (or momentum flux, friction velocity) was computed in Fig. 11? Based only on the longitudinal (or downstream) $< u'w' >$ wind stress component or both longitudinal and lateral (or crosswind) $< v'w' >$ stress components? Why?**

The referee is right when stating that the data processing and corrections were briefly mentioned. Following his/her query, we have included further information about the filtering, data quality control checks and correction of the turbulent fluxes, as well as minor manual checks, in the new manuscript (Page 5 Lines 22-34, and Page 6 Lines 1-6).
Regarding the calculation of the friction velocity, since the double rotation (and not the triple rotation) has been applied to the sonic coordinate system (as specified in the new manuscript), the friction velocity was calculated considering both the longitudinal- and lateral-stress components.

**3. Large errors in the measurement of the turbulent fluxes can result from relatively small errors in the alignment of a sonic anemometer due to the cross contamination of velocities (i.e. fluctuations in the longitudinal wind speed components appear as vertical velocity fluctuations, and vice versa). To avoid these**

errors rotation of the anemometers' axes is needed to place the measured wind components in a streamline coordinate system. The most common method is a double rotation of the anemometer coordinate system to compute the longitudinal, lateral, and vertical velocity components (Kaimal and Finnigan, 1994, section 6.6). Was this done?

We agree that rotation of the sonic-anemometer axes is needed to prevent errors in the estimation of the turbulent parameters. Indeed, the double-rotation method was applied to compute the longitudinal, lateral and vertical velocity components. It has now been specified in the new manuscript (Lines 32-34 on Page 5).

**4. Since the sonic anemometer measures the so-called 'sonic' virtual temperature (which is close to the virtual temperature) the moisture correction in the sonic anemometer signal is necessary to obtain the correct value of temperature itself and sensible heat flux (e.g. Kaimal and Finnigan, 1994). Authors reported the sensible heat flux (Figure 9). To value the present results the authors should either show that the moisture corrections and their impact on the results are small, or (if otherwise) apply moisture corrections to the sonic temperature following Schotanus et al. (1983) based on the data collected by the Campbell fast-response open path infrared gas analyzer listed in Table 1.**

Moisture corrections were applied to the sonic temperature following Schotanus et al. (1983) to derive the air temperature and sensible-heat flux. In fact, all the parameters based on the fast-temperature measurements shown in the figures along the manuscript have undergone this correction. It is described in the Easyflux DL software (Campbell Scientific, 2017) and now explained in the new manuscript (Line 34 on Page 5 to Line 2 on Page 6).
The effect of applying both moisture and Webb corrections (linked with next query by Referee#1) to the calculation of respectively the sensible and latent heat fluxes, is shown below in Fig. I for a selected downslope event within the analysed period.

*References:*

*Cambell Scientific: EasyFlux DL Eddy-Covariance CR3000 Datalogger Program,* `https://s.campbellsci.com/documents/us/product-brochures/b_easyflux-dl.pdf`*, 2017*

*Schotanus P., Nieuwstadt F.T.M., De Bruin H.A.R. (1983) Temperature measurement with a sonic anemometer and its application to heat and moisture fluxes. Boundary-Layer Meteorol. 26(1): 81–93. DOI:*`10.1007/BF00164332`
*.*

**5. Authors say nothing about the Webb correction (also referred as WPL or Webb effect after the paper by Webb et al. [1980]). This correction must be taken into account when the turbulent fluxes of minor constituents such as carbon dioxide or, in some cases, water vapor are measured (Webb et al. 1980).**

The referee is right that nothing about the WPL correction was mentioned in the manuscript,

[Figure]

Figure I: Buoyancy flux, sensible heat flux (after moisture correction) and the latent-heat flux before and after the Webb correction on July 19 2017 in La Herrería.

although it had been considered to correct latent-heat and $CO_2$ turbulent fluxes. It is mentioned now in the new manuscript (Line 2 on Page 6) so that the reader will know that this correction has been applied. As an illustrative proof, Fig. II compares the $CO_2$ turbulent fluxes before and after applying this correction for the same downslope event from Fig. I. It can be observed how the Webb correction is considerably important during daytime.

**6. In a slope-following coordinate system, the horizontal (along the slope) heat (buoyancy) flux contributes to the net buoyancy term and, therefore, the Monin-Obukhov stability parameter z/L (see page 11 and Fig. 10) contains this additional term (e.g., see Grachev et al. 2016, their Eq. (3) and references therein). Authors say nothing about this issue for calculation z/L which is very important point for katabatic flows.**

We thank the referee for this interesting and important aspect, but given the issues with the horizontal heat fluxes, the challenging application of the MOST theory stressed by Referee#2, and the fact that the calculated stability parameters are not strictly needed for the conclusions drawn in this study, we have removed former Fig. 10 from the old manuscript.

**Minor and editorial/technical comments:**

**I. Page 5, Line 8. Replace $CO^2$ by $CO_2$.**

[Figure]

Figure II: $CO_2$ turbulent fluxes before and after the WPL correction on July 19 2017 in La Herrería.

Corrected, thank you.

**II. Page 11, Line 28. I suggest to provide a definition of the Monin-Obukhov stability parameter (z/L) and the bulk Richardson number ($R_B$) for a layman reader.**

We thank Referee#1 for this suggestion, but for the reasons given in the response to Query 6 above, those definitions are not necessary anymore in the new version.

**III. Page 12, Lines 2-9. I would like also to see here a discussion on difficulties and controversy of interpretation associated with the critical Richardson number (e.g., Grachev et al. 2013 and references therein).**

For the reasons explained in the preceding comment, that discussion is not needed anymore.

**References: Replace 'Boundary Layer Meteorol.' by 'Boundary-Layer Meteorol.' (dash is missed).**

Corrected, thank you.

**Page 19, Line 1. Please correct Silvana's name: Di Sabatino S. instead Sabatino, S.D.**

Corrected, thank you.

**Page 20, Line 9. Please correct reference Pardyjak et al. as follows: Pardyjak E.R., Fernando H.J.S., Hunt J.C.R, Grachev A.A., Anderson J.A. (2009) A case study of the development of nocturnal slope flows in a wide open valley and associated air quality implications. Meteorologische Zeitschrift, 18(1), 85–100. DOI: 10.1127/0941-2948/2009/362**

Accordingly corrected, thank you.

---

## Author Comment (AC2) · 21 Mar 2019

**Responses to comments from Referee #2**

**MANUSCRIPT: acp-2018-944**
**TITLE:** From weak to intense downslope winds: origin, interaction
with boundary-layer turbulence and impact on $CO_2$ variability
**AUTHORS:** Jon A. Arrillaga, Carlos Yagüe, Carlos Román-Cascón,
Mariano Sastre, Maria Antonia Jiménez, Gregorio Maqueda
and Jordi Vilà-Guerau de Arellano

―――――――――――――――

**MAIN CHANGES IN THE MANUSCRIPT:**

o  Title.

o  Abstract and motivating aspects.

o  Denomination: katabatic → downslope.

o  Further information about data postprocessing in Sect. 2.2.

o  New Sect. 4 in the revised manuscript: analysis of the heat and momentum budgets, profiles and the estimation of the jet-maximum height for three representative events.

o  Summary and conclusions.

o  Appendix A (footprint estimation) and B (assessment of the thermal profile).

o  Removed figures (numbers from the old manuscript): Fig. 7, Fig. 10, Fig. 11 and Fig. 12.

o  Merged figures (numbers from the old manuscript): Figs. 4 and 5, Figs. 8 and 9.

o  New figures (numbers in the revised manuscript): Fig. 7, Fig. 8, Fig. 9, Fig. 10, Fig. A1 and Fig. B1.

o  Slightly modified figures (numbers in the revised manuscript): Fig. 1 and Fig. 11.

o  Wording and English review.

**General comments:**

The authors study downslope flows at a site close to the Guadarrama Mountain Range by means of mean and turbulence measurements at a 10 m tower. The authors use an algorithm to identify periods with low synoptic forcing and katabatic flows, and then separate the periods into those with weak, intermediate and strong katabatic flows. Finally, they study the conditions under which each of these occurs and the associated boundary layer structure and CO2. The study shows some interesting, though not surprising results on the correlation between the strong wind episodes that lead to weakly stable stratification, and weak wind episodes that lead to very stable stratification. Still, the study has major gaps and in general lacks a thorough analysis of physical processes and associated budgets needed to substantiate the explanations which are at the moment sometimes given without a thorough proof (see specific comments below on whether this flows are katabatic at all and what their origin is given that katabatic flows cannot develop in unstable stratification, on the need to look at budgets, or for example the text connected with Figure 3). Also, there is a lack of thorough understanding of the nature of these flows (both in terms of the driving mechanisms and the interaction with turbulence) and the fact that the location of the jet maximum is a vital information that is lacking from the entire study, if indeed this flows are shown to be katabatic. If the turbulence data are collected from above the jet maximum, then it is turbulence that is not connected with the ground and therefore is not expected to show standard boundary layer characteristics (such as MOST etc). Where the jet maximum within the tower depth for each averaging period is needs to be added in the discussion of all the results and all the discussions and conclusions adjusted accordingly. For a more in depth study of the turbulence characteristics of katabatic flows Grachev et al. (2016) paper gives excellent information.

We thank Referee #2 for his/her useful suggestions and comments about the manuscript. We have considered all the major points and modified the manuscript accordingly. First, and directly associated with the concern about the nature of katabatic flows in this study, we have decided to change the way we refer to them to the generic "downslope flows" (Zardi and Whiteman, 2013)). The algorithm ensures that they are to some extent thermally-driven, but since some of them have a dynamical input, we have decided to denominate them, as a whole, downslope flows. Some of them, in any case, behave as pure katabatic flows as it is indicated in the new manuscript. Moreover, the physical processes underlying the formation and development of the downslope flows have been more deeply explored. For that, the heat and momentum budgets have been calculated and investigated for the individual cases. Additionally, a moderate downslope flow has been added to the analysis of individual representative events. For these events, the interaction with turbulence, their dynamical and thermal characteristics, as well as the location of the jet maximum have been explored. Please notice that Sects. 4.2 and 5.1 have been eliminated from the old manuscript and new sections have been added in the new manuscript (Sect. 4, Appendix A and B). The responses to the specific queries from the referee, are provided point-by-point below. The modifications

undertaken in the manuscript can be checked up both from the revised manuscript and the tracked-changes version provided.

**Specific major comments:**

**1. Turbulence data processing. As already mentioned by the first referee, the authors fail to give vital information on the turbulence data processing. Apart from the missing information on rotation methods and turbulence corrections, the authors also fail to motivate why they use a 10min averaging time, which in stable boundary layer is generally too long, and even for strong wind conditions the more appropriate averaging time would be 5 min, while for weak winds it is most likely 1 min or less. I suggest the authors calculate ogives or multi-resolution flux decomposition (e.g., Vecenaj et al, 2012) for their 40 episodes and estimate the most appropriate averaging time. If they need to have 10 min averages for comparison to the slow sensors, then the fluxes can be afterwards averaged to that value.**

Referee#2 is right when pointing out that information about the corrections and data post-processing procedures was missing. This was also a query from Referee#1. As stated in the responses to Referee#1, this information has been included in the new manuscript (Lines 22-34 on Page 5 and Lines 1-6 on Page 6).

With regard to the appropriate averaging time, Multi-resolution flux decomposition (MRFD) of the friction velocity and kinematic heat flux for the three individual events have been calculated at both 4 and 8 m, to support the election of the 10-min window. They represent the distinct turbulent conditions that are found within the database of 40 events. We show MRFDs for the intense-downslope (27/07/2017), moderate-downslope (25/07/2017) and weak-downslope (13/08/2017) events only at 8 m, since the interpretation is very similar at 4 m.

The MRFD of both the friction velocity and kinematic heat flux for the intense event (Fig. I) show that the centre of the spectral gap is between 5 and 10 min. However, an averaging time of 10 min seems to be more appropriate than 5 min to capture all the turbulent scales. The MRFD of the friction velocity for the moderate event (Fig. IIa), shows the centre of the spectral gap also between 5 and 10 min. On the other hand, the spectral gap from the kinematic heat flux is unclear.

Finally for the weak downslope event, in which the uncertainty is greater, the centre of the spectral gap between 1600 and 1800 UTC is located between 5 and 10 min as for the two other events. If we choose 5 min as the averaging time we lose some scales, whereas when choosing 10 min we may include a few non-turbulent scales. After 1800 UTC turbulence is very weak and the spectral gap is hardly distinguished.

After exploring the MRFDs for the three events we can conclude that in general 10 min is the most appropriate averaging time in order to include all the turbulent scales for the distinct downslope times. Many studies have found that due to factors such as stability, mesoscale circulations and the synoptic forcing, the spectral gap can turn vague or highly variable (e.g. Hess and Clarke, 1973; Viana et al. 2010; Román-Cascón et al. 2015; Schalkwijk et al., 2015; Babic et al., 2017). In particular, when working with a large database, changing the

[Figure]

Figure I: *MRFD analysis of the friction velocity and (b) kinematic heat flux at 8 m for the intense downslope event (27/07/2017) between 1600 and 2330 UTC. Horizontal white lines indicate timescales of 1, 5 and 10 min.*

[Figure]

Figure II: *Same as Fig. I for the moderate downslope event (25/07/2017).*

averaging time by adapting it to different turbulent conditions is impractical and subjective, so it is preferable using a standard averaging time. 10 min is considered standard for micrometeorological datasets (Mauritsen et al., 2017).

*References:*

*Babić, N., Večenaj, Z., and De Wekker, S. F. J. (2017). Spectral gap characteristics in a daytime valley boundary layer, Q. J. R. Meteorol. Soc., 143, 2509–2523. DOI:* `https:`

[Figure]

Figure III: *Same as Fig. I for the weak downslope event (13/08/2017).*

*// doi. org/ 10. 1002/ qj. 3103 .*

*Hess, G. D. and Clarke, R. H. (1973). Time spectra and cross-spectra of kinetic energy in the planetary boundary layer, Q. J. R. Meteorol. Soc., 99, 130–153. DOI: https: // doi. org/ 10. 1002/ qj. 49709941912 .*

*Mauritsen, T. and Svensson, G. (2007). Observations of Stably Stratified Shear-Driven Atmospheric Turbulence at Low and High Richardson Numbers, J. Atmos. Sci., 64, 645–655. DOI: https: // doi. org/ 10. 1175/ JAS3856. 1 .*

*Román-Cascón, C., Yagüe, C., Mahrt, L., Sastre, M., Steeneveld, G.-J., Pardyjak, E., van de Boer, A., and Hartogensis, O. (2015). Interactions among drainage flows, gravity waves and turbulence: a BLLAST case study, Atmos. Chem. Phys., 15, 9031–9047. DOI: https://doi.org/10.5194/acp-15-9031-2015.*
*Schalkwijk, J., Jonker, H. J. J., Siebesma, A. P., and Bosveld, F. C. (2015). A Year-Long Large-Eddy Simulation of the Weather over Cabauw: An Overview, Mon. Wea. Rev., 143, 828–844. DOI: https: // doi. org/ 10. 1175/ MWR-D-14-00293. 1 .*

*Viana, S., Terradellas, E., and Yagüe, C. (2010). Analysis of Gravity Waves Generated at the Top of a Drainage Flow, J. Atmos. Sci., 67, 3949-3966. DOI: https: // doi. org/ 10. 1175/ MWR-D-14-00293. 1 .*

**2. Footprint analysis.** The authors themselves talk about the footprints influencing the values of the fluxes (Pg 4, ln 33 or Pg. 14, ln 23-24), however, no footprint analysis is provided. I suggest the authors use the footprint model of Kljun et al. (2015) to examine the differences in the source area of the turbulent fluxes for the three different categories. But on another note, the sentence on

**Pg. 4 is erroneous: the footprint does not induce "uncertainty in estimation of fluxes" if the fluxes are calculated from the eddy covariance, they might just represent turbulence originating from other locations.**

As suggested by the referee, we have added the footprint analysis in the new manuscript. It has been added in Appendix A. To estimate the footprint we have employed the approximate analytical model from Hsieh et al. (2000), which is based on lagrangian stochastic dispersion models and dimensional analysis, in combination with the 2D extension from Detto et al. (2006), to include the contribution of lateral spread. This footprint model was chosen because it is developed for thermally stratified surface layers, it is practical and has been applied in many studies, giving satisfactory results when compared with other footprint models. In order to use the model from Kljun et al. (2015), however, we would have to estimate the boundary-layer height. But, as pointed out by Referee#2, when having a low-level jet the turbulence above the jet maximum might be disassociated with the surface, and therefore the estimation of the boundary-layer height has a great uncertainty. Therefore, we have chosen the models from Hsieh et al. (2000) and Detto et al. (2006) instead of the one from Kljun et al. (2015).

With respect to the sentence from the old manuscript brought by the referee, we do agree it is erroneous. Hence, it has been eliminated from the new manuscript, and the explanation associated to the footprint analysis has been revised.

**3. Structure of the katabatic flow. I find myself wondering if not doubting if the flow the authors are studying should be classified as katabatic at all or not. Katabatic flows possess very specific characteristics: a low level jet, formation due to surface temperature deficit and retardation due to surface friction (turbulent momentum transport towards the surface), very specific turbulent structure associated with its jet profile: negative momentum and positive horizontal heat flux below the jet and the opposite above, minimum in TKE at the jet maximum (see Grachev et al. 2016). The profiles in Fig. 11, particularly for the strong cases do not resemble katabatic ones at all, and the weak cases have a low level maximum that could be also just due to the interpolation scheme, and then appears to have a secondary maximum above the height of the tower. On a side note: why are the sonic measurements not used in the wind speed profiles such as in Figure 11? Could it be that it actually is the basin flow (Pg 2, ln 15. How do you ascertain that your flow is not actually influenced by the Madrid basin and is purely katabatic). The authors should look at the profiles of the turbulence quantities to first identify if their flow qualifies as katabatic and second to actually show if their profiles in Fig. 11 are physical at all or the low level jet in the weak case is purely a construct of interpolation scheme, and what is happening with the secondary maximum. The profiles of turbulent quantities would also allow them to estimate the jet maximum height in each individual period. The jet maximum height is indeed the vital parameter when studying anything related to katabatic flows since at jet maximum wind speed will be maximal but the turbulence will be zero – and thus exactly the opposite of standard flat-terrain stable boundary layer structure that the authors so heav-**

**ily rely on in the HOST, MOST, shear capacity and other diagnostics. In that respect, if there is really a low level jet maximum below 3 m in the weak cases, then the turbulence above that height light be disassociated with the surface and therefore not exhibit standard boundary layer characteristics. Not taking this fact into account invalidates the conclusions.**

Regarding referee's first wonder, and as previously explained, we have changed the denomination of all the events as a whole to "downslope flows". As explained in Sect. 4 of the new manuscript, weak downslope flows share the characteristics of katabatic flows, but moderate and particularly intense downslope flows show a distinct behaviour due to the dynamical input from the nearby slope. We have explored the physical mechanisms responsible for their formation by calculating the heat and momentum budgets for the three representative events in Sect. 4.1. In addition, we have included the profiles of the turbulent fluxes in Figs. 8–10, which support the existence of the katabatic jets on the weak downslope events (rejecting therefore the idea about being a construct of the interpolation scheme), and of the jet maxima above 10 m particularly on the intense events. A deep analysis about the physical mechanisms underlying, the thermal structure of the flow, as well as their interaction with turbulence for a representative weak, moderate and intense downslope event has been included in Sect. 4.

With respect to the wind-speed measurements from the sonic anemometer, they have not been included in the profiles because they introduce an extra instrumental bias on the wind profile, which is based on the cup-anemometer measurements at 3, 6 and 10 m. In some weak events, wind speed is very weak at all levels ($\simeq 1$ m s$^{-1}$), and therefore the instrumental bias could introduce an important deviation from the real profile. An example of this instrumental bias for the investigated weak event (13/08/2017) is shown in the following Fig. IV. It can be observed that the form of the peaks and minima is not always coincident between both instruments. Since from the cup anemometers we cover a larger vertical profile, we use those measurements to characterise the wind profile.

With regard to the downbasin flow, it is easily discriminated from the local downslope flow in our study site. Downslope flows are directed from the Guadarrama mountain range (i.e. from the NW), whereas downbasin winds approximately follow the direction of the main rivers of the area (i.e. from the NE during night-time). This distinction was also made in Plaza et al. (1997). Despite some of the events are affected by the irruption of downbasin winds, in the figures we just represent the profiles at times in which downslope flows are blowing, as required by the selection algorithm. Fig. 1a in the new manuscript has been changed, so that local slopes and the basin are better distinguished.

And regarding the last comment about the connection of the measured turbulence with the surface, the analysis of the regime transition from non-dimensional parameters (Sect. 4.2 from the old manuscript) has been eliminated, and therefore MOST does not need to be assumed anymore. With regard to the figure with the HOST transition (Fig. 7b), we believe that the transition of nocturnal regimes explained in Sun et al. (2012) is clearly identified from the represented turbulence data at 8 m. Furthermore, Sun et al. (2012) relate the occurrence of the distinct turbulent regimes with the existence of low-level jets during CASES-99.

[Figure]

Figure IV: *Times series of the wind speed from 1600 to 2400 UTC on the 13/08/2017. Solid lines represent the measurements from cup anemometers, and dashed lines from sonic anemometers.*

**4. Study of budgets** The authors should present budget of the momentum and heat to substantiate their claims (such as the section 3 and 4 when talking about the development of the flow and its interaction with turbulence and transition to very or weakly stable boundary layer), and also to more fully understand the processes at hand. By examining the budgets of the katabatic flow one could isolate the importance of individual terms (local generation, dissipation, advection etc) on the weak, intermediate and strong katabatic flows and therefore show if the weak katabatic flow for example is locally driven and the strong katabatic flow is advected from the steep slopes 2km away, whereby the change in slope (from 25° to 2°) leads to the deepening of the flow as observed by Smith and Skyllingstaad (2005). The budgets will also show the importance of mesoscale and not just the large-scale pressure gradients on the flow, even if only as a residual term. The budgets could answer where the claims that stronger unstable turbulence facilitates intense katabatics. This indeed is counterintuitive as for the katabatic flow to develop one needs a large temperature deficit (i.e. cooling) and turbulence suppresses the katabatic flow while unstable stratification does not even allow the development of katabatic flow (Pg. 8, ln 7-13).

As requested by the referee, the analysis of the heat and momentum budgets has been included in the revised manuscript (Sect. 4.1). Indeed, we have been able to prove that weak downslope flows (which behave as pure katabatic flows) are driven by the buoyancy acceleration triggered by the local temperature deficit, whereas moderate and particularly intense downslope flows, have an important dynamical contribution from the nearby steep slope. Hence, the arrival of the downslope flow can take place before the thermal profile becomes stable at the measuring point, and the strengthening of turbulence is not constrained by negative buoyancy.

**5. Stratification How was the virtual potential temperature calculated? Was the humidity needed to convert air temperature to virtual potential temperature used from Irgason and at which level? Also about the calculation of the potential temperature gradient: On Pg. 7, ln 20 and Equation 1, if you are using a 3th order polynomial why is the stratification calculated only from delta? The true temperature gradient dTheta/dz (if one takes the derivative of Eq 1) has contributions from beta, gamma and delta and depends on height.**

As explained in Lines 22-23 on Page 8, virtual potential temperature was calculated from temperature measurements from aspirated thermometers at 3, 6 and 10 m, and humidity measurements in a T/RH probe at 2 m.

With regard to the logarithmic fit of the virtual potential temperature profile, even though $\theta_v$ also depends on $\beta$ and $\gamma$, the static stability is best described from the value of $\delta$. This information has actually been extended and included in Appendix B, by showing an example of the distinct static stabilities of the thermal profiles during a moderate event. We did not mean to say that the gradient is independent of $\beta$ and $\gamma$, but that the static stability of the profile can easily be inferred from this parameter. The classification from Eq. B2 was designed after a thorough check.

**6. Origins of the flow. Tied to the previous comments, the paper fails to determine the origin of the katabatic flows. For example, on Pg. 8, ln 5 the authors mention that the stratification is unstable and net radiation positive during the onset of the strong katabatic episodes. Given that the katabatic flow is caused by stable stratification (temperature deficit) and therefore cannot develop in unstable stratification, the authors should show evidence of why they think their flow is katabatic, and how and where it originates from (does it originate on the steep slope where the stratification has already turned stable due to shadowing? And is now merely advected to the study site?)**

As explained in previous responses to queries, the investigation of the budgets has provided further proof about the origin of the distinct downslope flows: weak downslope being katabatic, and moderate and intense downslope being partly or mostly dynamically induced by the nearby slope. Considering that the axis of the mountain range is directed SW-NE, sunset takes place at the back of it, and therefore the cooling down of the surface starts earlier in the slope than at the foothill. If the low soil moisture favours the stronger cooling and the synoptic wind, despite being weak, is from the NW, the downslope advection of cold drainage fronts is possible, arriving at the foothill when still the thermal profile is unstable and net radiation positive. Such scenario was for instance observed in Papadopoulos and Helmis (1999).

**7. Figure 7 and the correlation to soil moisture Putting the 4th order fit through the data presented in Figure 7 is stretching it beyond any justifiability. Indeed, the spread of the data is so large that a linear fit would be possible at maximum to show that for low soil moisture G is slightly positive, but for high it is mostly negative. The results for longwave-radiative loss show no correlation between the data, both linear or non-linear. I therefore protest against any conclusions**

**based on these fits and the identified two maxima.**

We agree with the referee that the spread of the data is large and that the correlation is low. Therefore, we have removed the figure and the associated explanation from the manuscript. Instead, a short comment about this vague correlation has been included in the revised manuscript (Lines 21-24 on Page 9).

**8. Energy balance closure and the ground heat flux. It is interesting to note that the level of energy balance closure is so high in the study site. How did the authors calculate the ground heat flux and the heat storage in the layer above the heat flux plate? Given the nature of the weak and the strong flows, one could also argue on the importance of advective processes, but the energy balance appears to suggest that advection is not important for strong katabatic flows.**

Given the uncertainty in the calculation of the energy-balance closure, the figure in which we showed the different components has been eliminated. Instead, we have preferred to explore the nature of different downslope flows by analysing the heat and momentum budgets, as suggested by this referee. In fact, the analysis of the energy-balance closure is out of the scope of this study.

**9. Interaction with turbulence The second motivation of the study talks about the interaction between turbulence in SBL and katabatics but actually, the relevant turbulence that is interacting was found to be unstable stratification before the onset of the wind itself.**

The referee is right when stating that not just turbulence in the SBL is interacting with downslope flows. Indeed, the moderate and intense downslope flows that are taken as case studies, are established before the onset of the SBL. For that reason, the second motivating aspect of the paper has been changed from "The interaction of katabatic winds with local turbulence in the SBL and the implication in turbulent characteristics" to "The interaction of downslope winds with local turbulence and the implication in the characteristics of the SBL". Note that we still mention the SBL, since the characteristics of the SBL (once it is established) are strongly affected by the nature of the different downslope flows, and is one of the motivating aspects of the study. On the other hand, the term "stable boundary layer" has been replaced by "boundary layer".

**10. Language I suggest a professional or native speaker to check the language as I didn't want to enumerate all the things that need to be changed (e.g. "avoids" should be "prevents", "striked" should be "stricken" or rather something more appropriate, "fogs" as plural does not exist, it is always "fog", "emplacement" sounds awkward etc.)**

As suggested by Referee#2 and Referee#3, a native speaker has revised the manuscript and given some language corrections and improvements, apart from those provided by the

referees.

**Individual Major Corrections**

**2. Pg. 1, ln 1. What are the "dynamic and turbulent" features of SBL?**

We have changed it to "the turbulent characteristics and thermal structure".

**3. Pg. 1, ln 8: In Figure 6, the limits is 3.5 m/s not 6. What is the correct number?**

The lower limit is 3.5 and the upper limit 6: it has been clarified in the new manuscript.

**4. Pg. 3, ln 8-14: No, even over flat terrain with the existence of a low level jet or even in very stable stratification without a low level jet MOST is not valid**

As commented in the response of Query 6 from Referee #1, Fig. 10 from the old manuscript was eliminated for various reasons. Therefore, the validity of MOST is not assumed anymore in this analysis.

**5. Pg 4, ln 24: why would a strong surface thermal inversion necessarily be allowed to develop just because the slope angle is low, if there is enough wind to prevent its development?**

This referee is right when stating that strong wind can prevent the development of the thermal inversion. However, what we mean is that over a shallow slope the buoyancy acceleration is smaller than over a steep slope, and therefore the thermal inversion can be stronger, just with the locally generated flow and without the erosion of drained downslope flows from the nearby steep slope. This has been clarified now.

**6. Pg. 4, ln 35: you have not mentioned the CO2 fluxes at all until the moment when you mention the negligible effect of the urban area. Or do you mean the effect of urban area on all the fluxes?**

We meant the effect on all the turbulent fluxes. In any case, the explanation associated with the footprint has been changed in the new manuscript (Lines 4-8 on Page 5 and Appendix A).

**7. Figure 3. Why is there no data from the sonics? Also, the way the data are presented all lumped together does not show if there are wind maxima at different heights for the different episodes, periods. I suggest the authors calculate the jet maximum if possible, normalize the height with it and then plot the normalized profiles.**

With respect to the wind data from the sonic anemometers, as explained in the Specific Major Comment 3, they have not been included in the profiles or Fig. 3 due to the instrumental

bias they introduce. Apart from that bias, due to the closeness between the levels and to the fact that sonic levels are in between cup-anemometer measurements, they do not provide further information of interest about the wind profile, as can be seen from the following Fig. V.

[Figure]

Figure V: *Same as Fig. 3a from the manuscript but including sonic measurements at 4 and 8 m.*

On the other hand, the referee is right that this plot does not show the location of wind maxima. In fact, as inferred from the sign of the turbulent fluxes in Figs. 8–10, the jet maximum in many cases is probably located above 10 m. This plot was included in order to show the wind-speed frequency distributions and range of values at different levels, and to introduce and motivate the classification of events. In addition, as shown for the three representative events in Sect. 4.2, the determination of the jet-maximum height is not always possible only with two sonic measurements. Therefore, even though the normalisation of the height with the jet-maximum location is an interesting exercise, it is not feasible in this study without carrying great uncertainty.

**8. Pg. 6, ln 20-26: Nowhere in Figure 3 is it visible that there is a development of skin flow or where the jet maximum is. Indeed, the figures seem to suggest that the jet maximum is always above 10 m.**

We agree that from Fig. 3 the development of a skin flow and the position of the jet maximum are not visible. We only state that the median at 3 m is similar to that at 6 m, and the first quartile is even smaller at 6 m than at 3 m, which does not occur for the third quartile. This occurs as a consequence of the skin flow on the weak downslope events, but not for all the downslope events, and therefore it is not observed for the whole distribution. The skin flow is shown in Fig. 10b (weak event), and is clearly absent in Fig. 12b (intense event),

since the jet maximum is above 10 m.

**9. Figure 3 should also include the information on the temperature profiles and turbulence.**

We acknowledge this suggestion but we prefer just to include the box plots for the wind speed. At the point of the manuscript where this Fig. 3 is introduced, frequency distributions for the temperature and turbulence would not contribute to the motivation of the classification into the three downslope events, which is the main purpose of this figure. On the other hand, thermal and turbulence profiles are shown in Figs. 8–10, and we think they provide more interesting information on the thermal structure and interaction with turbulence than the frequency distribution of these variables itself.

**10. Pg 7, ln 6-7: I find it very strange that the results at 3 and 10 m do not show conformation to the classification and only 6m is so good. The sonic data should be used to study this more in detail and give a physical explanation why this is so.**

In those lines (Lines 10-11 on Page 8 of the new manuscript) we state the following: "At the levels of 3 and 10 m the events showing different features cannot be so clearly detached, and therefore the level of 6 m is employed for the classification". We did not mean that the levels of 3 and 10 m do not show conformation to the classification, but that the classification into the three downslope types with contrasting turbulence conditions is better produced using the level of 6 m. On the other hand, as commented above, due to the instrumental bias we prefer not to use measurements from sonic anemometers, but from cup anemometers.

**11. Flocas et al. reference is missing from the list of references**

Thank you for pointing out the missing reference. It has been included in the revised manuscript.

**12. Pg. 7, ln 17: is it the soil or the skin temperature?**

It is actually the skin temperature. It has been corrected.

**13. Pg. 8, ln 1: is the low value of TKE due to the jet maximum being close to the measurement height or because one is above the jet?**

In some cases it could be due to the closeness of the jet maximum, but in that group some moderate downslope flows are also included, and the skin flow is not always developed. With the available information, we cannot specify the position of the jet maximum for all the events within the group of low TKE. Note that Figs. 4 and 5 from the old manuscript have been merged into Fig. 4 of the new manuscript.

**14. Pg 8, ln 6: katabatic flow develops turbulence through shear generation. What you mean to say by "relation between katabatic flow and turbulence" I**

**guess is, the turbulence before the onset of the flow**

We think that the interaction is bidirectional. The turbulence at the onset influences the downslope flow, and at the same time, the downslope flow itself affects the turbulent characteristics of the boundary layer. This can be checked up for instance from Figs. 8–10.

**15. Pg. 9, ln 3: why would large soil moisture after precipitation during nighttime lead to the enhanced cooling of the soil?**

From the available data, we cannot draw conclusions about the mechanism explaining that fact without being speculative. Additionally, the figure associated with this explanation has been eliminated from the manuscript. Hence, investigating the mechanism that explains why after strong precipitation surface cooling can be enhanced is not within the scope of the paper.

**16. Pg 9, 7-13: The study on the influence of stratification needs to be associated with momentum and heat budgets of the flow to show whether the conclusions drawn are substantiated in the text.**

Now, the heat and momentum budgets have been calculated and analysed in Sect. 4.1.

**17. Pg. 10, ln 8-19: the transition will depend on the location of the jet maximum and if it is below 6m or above or is moving between. The exact value that is lower than in Sun is indeed no wonder given the fact that there is a jet maximum present and not in Cases-99.**

As explained in the Specific Major Comment 3, Sun et al. (2012) relate the occurrence of the distinct turbulent regimes with the existence of low-level jets during CASES-99. In our case, the transition occurs independently of whether the jet is located below or above the sonic measurements. Even if the skin flow is present (below 3 m) for many weak downslope events, we cannot assure that the jet maximum is close to the sonic measurements whenever $U < 1.5$ m s$^{-1}$ (including some moderate events in black). We can only comment about this possibility (sentence included now on Page 11 Lines 3-4).

**18. Pg. 11, ln 1-3. The two sentences on MOST cannot be applied to the current study without more understanding of the processes studied. MOST will only be applicable very close to the surface if the stratification is weakly stable and turbulence is well developed, and if the terrain is horizontally homogeneous. If there is a low level jet close to the tower height or even worse, within the tower height, MOST by definition is not valid as there is another height scale that is more important than z/L.**

For the reasons presented in the response to Major Point 6 from Referee#1, Fig. 10 from the old manuscript has been eliminated. Therefore, the compliance of MOST is not assumed anymore in this work.

**19. Pg. 11, ln 25: Isn't the fact that Fig 10b resembles Fig. 8b by construction since you change the definition of shear capacity to match the HOST?**

Idem to the previous response, Fig. 10 has been eliminated.

**20. Pg 12, ln 4-6: The calculation on Rb will again depend on the existence of the jet maximum if it is below, and therefore it doesn't make much sense as a measure. A better measure would be the gradient Richardson number Ri which the authors could calculate from the interpolated profiles and therefore obtain a profile of Ri.**

Idem to the previous response, Fig. 10 has been eliminated.

**21. Pg 12, ln 24: say that the value of the diurnal peak is not shown, or do you refer to the little part before the transition shown in Fig. 11?**

We refered to the diurnal peak throughout the day, so we include "(not shown)" in the manuscript.

**22. Pg. 12, ln 25: does the flow arrive or develop? Show the budgets**

This is an important and interesting point that has been clarified in the new manuscript after showing the budgets. As it has been demonstrated, the weak downslope events are mainly locally generated katabatic flows, whereas some moderate and particularly intense downslope flows are dynamically induced and propagated from the nearby steep slope, so that we can say that the flow "arrives" for them.

**23. Pg. 12, ln 1-2: how accurate is this wind maximum that does not exist in the measurements but only in the interpolation?**

From the measurements, particularly at 2230 UTC, the existence of the jet maximum around or below 3 m can be guessed ($U_{3m} > U_{6m}, U_{10m}$; Fig. 10b). Moreover, the profiles of the momentum and horizontal-heat turbulent fluxes (Fig. 10d) support the existence of the jet below 4 m.

**24. Pg. 12, 5-6: In Grachev et al. (2016) paper it says that the flow is stationary but only in the well-developed phase. You are focusing on the transition.**

Thank you for pointing this out. That sentence has been eliminated from the manuscript.

**25. Pg. 15, ln 23: "form when maximum wind speed is kept" should be "have maximum wind speeds below", because indeed you are talking about the katabatic wind speed of 1.5 and not the ambient wind speed into which the katabatic wind impinges.**

Thank you for the suggestion, it has been accordingly changed.

**26. Pg. 15, 26: Wind shear is not driving katabatic flow, the driver is the negative buoyancy and wind shear is the product of the katabatic flow itself.**

We do agree with the referee. That sentence has been eliminated and the whole explanation has been revised.

**27. Pg. 15, ln 29: "intense katabatics are found" should be "int. kat. have maximum wind speeds..". It is again a question of cause and effect.**

Thank you for the suggestion, it has been accordingly changed.

**28. Pg. 16, ln 10: the scaling regime expected to be valid for at least very stable conditions is local or most likely z-less scaling.**

Idem from Specific Major Comments 19 and 20. Fig. 10 from the old manuscript and the associated conclusions have been withdrawn.

**29. Pg. 16, ln 25: influence of submesoscale phenomena will be visible in the calculated ogives or multi-resolution flux decomposition.**

The referee is right that the influence of submeso motions is visible from the calculated MRFD plots (Figs. I-III), particularly over a timescale of around $10^3$ s. In any case, the analysis of submeso phenomena was not within the scope of the article, and therefore, MRFD analyses have not been included in the revised manuscript.

---

## Author Comment (AC3) · 21 Mar 2019

**Responses to comments from Referee #3**

**MANUSCRIPT: acp-2018-944**

**TITLE:** From weak to intense downslope winds: origin, interaction
with boundary-layer turbulence and impact on CO2 variability

**AUTHORS:** Jon A. Arrillaga, Carlos Yagüe, Carlos Román-Cascón,
Mariano Sastre, Maria Antonia Jiménez, Gregorio Maqueda
and Jordi Vilà-Guerau de Arellano

———————————

**MAIN CHANGES IN THE MANUSCRIPT:**

o  Title.

o  Abstract and motivating aspects.

o  Denomination: katabatic $\rightarrow$ downslope.

o  Further information about data postprocessing in Sect. 2.2.

o  New Sect. 4 in the revised manuscript: analysis of the heat and momentum budgets, profiles and the estimation of the jet-maximum height for three representative events.

o  Summary and conclusions.

o  Appendix A (footprint estimation) and B (assessment of the thermal profile).

o  Removed figures (numbers from the old manuscript): Fig. 7, Fig. 10, Fig. 11 and Fig. 12.

o  Merged figures (numbers from the old manuscript): Figs. 4 and 5, Figs. 8 and 9.

o  New figures (numbers in the revised manuscript): Fig. 7, Fig. 8, Fig. 9, Fig. 10, Fig. A1 and Fig. B1.

o  Slightly modified figures (numbers in the revised manuscript): Fig. 1 and Fig. 11.

o  Wording and English revision.

**General Comments:**
The manuscript investigates katabatic flows on the basis of occurrences observed during one summer season at the foothills of the Guadarrama Mountain Range in Spain. The data set has been split up into weak, moderate and intense events, based on the observed maximum wind speed observed during each individual case and is then analyzed under various aspects. The study shows distinc differences between the different classes of katabatic flow, the number of intensive katabatic flow cases is, however, very low (3) and rises thus questions on the statistical significance of the reported results. The paper is in general well structured and includes a good literature overview on the subject. The figure layout is in general rather inhomogeneous over the paper and should be reworked. Several of the figures are in addition hard to read, mainly due to small labels and legends. Finally, the manuscript requires a thorough makeover by a native English speaker. Main points in this context the rather complicated sentence structures, unconventional wording obviously taken from the dictionary (e.g. emplacement instead of site/location), missing commas, the improper use of prepositions and articles, and grammatically incomplete sentences. I have marked a quite a few, but far from all, instances in my specific comments.

We thank Referee #3 for his/her comments about the article. We agree that the number of intense flows is small, but anyway, some of the most important conclusions about the nature and characteristics of the downslope flows (note that we have changed their denomination from katabatic to downslope), are drawn from the analysis of representative individual events and not from the statistical parameters of the distributions. In any case, this new database could be enlarged in time by including further measurements. Furthermore, we have revised the figures and modified them in order to be legible and clear enough. Finally, a native speaker has revised English along the manuscript, providing language corrections and improvements, apart from those given by the referees. It must be noted that Sect. 4.2 and 5.1 have been eliminated from the old manuscript and new sections have been added in the new manuscript (Sect. 4, Appendix A and B). Below, we provide point-by-point responses to the specific queries from the referee. The modifications undertaken in the manuscript can be checked up both from the revised manuscript and the tracked-changes version.

**Specific comments:**

**1) P1, L1: "on the dynamics" instead of ""in the dynamics"**

We thank Referee#3 for all the language corrections and improvements suggested. The manuscript has been accordingly corrected.

**2) P1, L5: insert comma after "moderate and intense"**

Inserted.

**3) P1, L9: insert "flow" after "katabatic"**

Inserted.

**4) P2, L26: "at contrasting" instead of "in contrasting"**

Changed.

**5) P2, L28: insert comma after "model"**

Inserted.

**6) P2, L33: "In contrary" instead of "At contrary"**

Changed.

**7) P3, L8 and 11: "emplacement" is rather uncommon, better "site or location"?**

It has been changed to both location and site.

**8) P3, L15-16: "on the concentration" instead of "in the concentration"**

Changed.

**9) P3, L16: remove "the" before "CO2**

Changed.

**10) P3, L16: "in coastal areas" instead of "at local areas"**

Changed.

**11) P3, L27: sentence incomplete; "the role of: ......, in CO2 mixing ratios" ; should be "in controlling/affecting CO2 mixing ratios"**

Completed.

**12) P3, L34: insert "concentrations" after "CO2"**

Inserted.

**13) P4, L23: "a relatively" instead of "an relatively"**

Changed.

**14) P4, L23-24: "immediately besides" is quite strange; better "close to"?**

It has been changed to "close to".

**15) P4, L29-30: "needleleaved evergreen tree cover", sounds complicated; isn't it just "coniferous"?**

This name was obtained from the database and maps of land-use types from Land Cover CCI from ESA. In any case, in order to clarify, we have added "coniferous" in brackets.

**16) P4, L34: replace "inexistent" by "absence of"**

As the native speaker suggested, it has been changed to "non-existent".

**17) P5, L8: formatting error "CO2" subscript**

Changed.

**18) P6, L5: insert "concentrations" after "CO2"**

Inserted.

**19) P6, L7: "Forty were selected as days: : :: : :.." The sentence is grammatically poorly formulated and hard to read. Please rephrase**

It has been changed to "Forty events with the formation of a thermally-induced downslope flow were selected from the analysed summer period with available data (94 days in total)".

**20) P7, L2: insert "the criteria" after "meet"**

It has been accordingly changed.

**21) P7, L2 (and other instances): replace "minutal" by "minute"**

Replaced.

**22) P7, L4: replace "weak" by "low"**

Replaced.

**23) P7, L29: there is no Figure 5a)**

The former Fig. 5 has been eliminated, so this sentence is not present anymore.

**24) P8, L34: Why are you presenting a 4th-order polynomial fit; any physical reasoning for this? A simple trend could also be seen from a linear regression, and the two peaks resulting from your fit seem to be rather arbitrary; Thus I see a big danger of an over-interpretation of non-existing features in the corresponding paragraphs on page 9, L1-13 (see also my comment on Figure 7)**

As commented in the response to the Specific Major Comment 7 from Referee#2, the reason for using the fourth-order fit was to justify the bimodal behaviour. However, due to the vague relationship between the variables presented, that figure and the associated explanation have been removed from the manuscript. A short comment about this vague correlation has been included in the revised manuscript (Lines 21-24 on Page 9).

**25) P9, L17-18: "......, the shear associated with the katabatic flow increases, and the downslope flow strengthens progressively." I do not understand the direct link between this two statements; How can increase in shear strengthen the downslope flow? Might also be a misunderstanding from my side, but then the sentence should be rephrased.**

That sentence has been changed to "If the downslope flow arrives when the stratification is still unstable and the surface thermal inversion (hereinafter measured from $\theta_v$) is not formed yet, the downslope flow strengthens progressively" (Page 9 Lines 33-34).

**26) P9, L20: insert "the" before "surface"**

Inserted.

**27) You should define the calculation of VTKE already here, and not two lines under.**

As suggested by the referee, it is defined two lines above.

**28) P10, L2: insert comma after the parenthesis with the wind speed.**

Inserted.

**29) P10, L10: "increases linearly"; I could also see a square root dependency here**

Given the wonder from Referee#3, we have represented in green the linear fit (Fig. Ia) and the square-root fit (Fig. Ib) over the scatter plot of $V_{TKE}$ vs $U$ when $U > 1.5$ m s$^{-1}$, together with the value of $r^2$. The correlation is very similar for both plots (the rounded $r^2$ is equal), indicating that it is not clear whether the dependency is linear or square root. Therefore, we have changed the mentioned sentence to: "increases approximately at a linear rate with $U$" (Line 28 on Page 10).

[Figure]

Figure I: *Same as Fig. 7b from the manuscript including in green (a) the linear fit, (b) the square-root fit, and the value of the square of the multiple correlation coefficient ($r^2$) for the fit .*

**30) P10, L29: replace "on" by "for" or "during"**

That sentence is eliminated from the revised manuscript.

**31) P10, L32: insert comma after "night"**

Inserted.

**32) P11, L17: Sentence has to start with an upper case letter; "Van Hooijdonk: : :..."**

Corrected.

**33) P11, L25-26: "intense and weak katabatics cluster into two clearly distinct regimes"; at 8 m I still see a considerable amount of black and red data points for SC¡3 with distinct elevated VTKE levels; can you explain/comment on this.**

We thank the referee for this comment but Fig. 10 has been eliminated from the manuscript, as well as the associated text.

**34) P12, L4: put "by definition" between commas.**

That sentence has been eliminated from the manuscript.

**35) P12, L17: replace "related with" by "related to"**

Replaced.

**36) P13, L7-8: start the sentence with "In contrary, U remains: : :.."**

Changed.

**37) P13, L12: insert comma after "takes place"**

Inserted.

**38) P13, L25: replace "so doing" by "doing so"**

Changed.

**39) P14, L9: insert comma after "SBL"**

Inserted.

**40) P14, L20: insert comma after "nearly 0"**

Inserted.

**41) P14, L21: insert comma after "assumption"**

Inserted.

**42) P14, L22: insert comma after the reference**

The comma has been inserted before "following the methodology", since we think it is better for the meaning we want to express.

**43) P14, L25: insert comma after "equation"**

Inserted.

**44) P 15, L8-10: this sentence has to be rephrased, maybe even better split in to or three! In particular complicated is the part ": : :by the presence upwind of a land use component of forest: : :: : :: : :"**

That sentence has been changed to: "This positive $CO_2$ advection is probably induced by the presence of a land use composed of forest, mosaic trees and shrubs upwind, towards the downslope direction. Given the increased plant respiration and soil flux, greater $CO_2$ concentrations are accumulated close to the surface during the night" (Lines 8-10 on Page 17).

**45) Some small inconsistencies in the references a. Boundary Layer Meteorol. vs Boundary-Layer Meteorol.; I believe the latter one is the usual b. A few journal abbreviations are not terminated by a period; e.g. Borge et al. and Plaza et al. c. Presentation of doi or not for articles**

We thank the referee for pointing out these inconsistencies. Journal abbreviations and citations along the manuscript have been revised and accordingly corrected. The DOI of all articles has additionally been included.

**46) Page 22, Table 1; inaccurate caption, I suggest: "Specification of the values measured and the devices : : :: : :"; the specification of a value is not technical!**

Changed.

**47) Page 24, Table 3; add number of occurrences for each class in the table; maybe also an idea to place the measurement frequency directly in the table for each sensor instead of using the footnote solution.**

As suggested by the referee, the number of occurrences of each type are included in Table 3, and the sampling rate is also included in Table 1.

**48) Page 25, Figure 1a; the location names are difficult to read; use larger fonts and bold style; in addition have the degree symbols in the axes labels an underscore that should be removed.**

The figure has been accordingly modified.

**49) Page 27, Figure 3: I suggest to split this figure in 3 separate ones for the weak, moderate and intensive cases; as it is presented now you loose a lot of information by the averaging; I would also like to see the 40 individual profiles in this plots, e.g. as grey lines in the background.**

We thank the referee for his/her suggestions about this figure. However, we prefer to keep it as it is for various reasons. First, one of the main purposes for including this figure is to represent the frequency distribution of the wind speed, show the differences between the levels at the onset and when the maximum intensity is achieved, and motivate the classification into the three types. Therefore, we think that at this point of the manuscript it is preferable to keep this figure as it is. Second, if we plot the 40 individual profiles in the background, the figure becomes fuzzy and unclear. Instead, we plotted the individual profiles for three representative events in Figs. 8–10. In this way, we compare in a clear way the structure of the downslope flows for the three types.

**50) Page 31, Figure 7: I cannot see that the applied 4th-order fitting makes any sense; do you have any physical reasoning for your choice.**

Please, see response to Comment 24 above.

**51) Page 34, Figure 10: labels/legends too small**.

This figure has been withdrawn from the manuscript.

**52) Page 35, Figure 11: a) use different line styles for 4 and 8 m (in particular important for the intense event in red); why have you selected 21:00 as last time you present; from the time series it looks like that is more a transition phase, while e.g. 22:00 appears to be a more stationary situation; b), d), f): I suggest to use a common x-axis span at least for the wind speed**

Thank you for these suggestions. This figure has been split into Figs. 8, 9 and 10 in the new manuscript. Different line styles are used for the different times represented. On the other hand, 2100 has been changed to 2230 UTC to represent a moment in which the SBL is already well developed. Besides, a common x-axis is used for all the panels in Figs. 8–10.

**53) Page 36, Figure 12: axis labels too small!**

This figure has also been eliminated from the manuscript: the evolution of the thermal stratification is shown in Figs. 8–10a; the time series of the wind shear is not needed either given that the friction velocity is also represented in Figs. 8–10a; and finally, the surface energy balance involves a great uncertainty, and hence, its interpretation is challenging.

**54) Page 37, Figure 13: use different line styles for 4 m and 8 m**

To better distinguish the two levels, instead of using different line styles, the level of 4 m is represented with the markers filled and the level of 8 m with the markers empty.